# Development of Pectin-Based Films with Encapsulated Lemon Essential Oil for Active Food Packaging: Improved Antioxidant Activity and Biodegradation

**DOI:** 10.3390/foods14030353

**Published:** 2025-01-22

**Authors:** Belkis Akachat, Louiza Himed, Merniz Salah, Maria D’Elia, Luca Rastrelli, Malika Barkat

**Affiliations:** 1Laboratory of Biotechnology and Food Quality (BIOQUAL), Institute of Nutrition, Food and Agro-Food Technologies (INATAA), Freres Mentouri University 1, Constantine 25000, Algeria; 2Institute of Industrial Hygiene and Safety, University Batna 2, Batna 05078, Algeria; 3National Biodiversity Future Center (NBFC), 90133 Palermo, Italy; mdelia@unisa.it (M.D.); rastrelli@unisa.it (L.R.); 4Department of Pharmacy, University of Salerno, Via Giovanni Paolo II, 132, 84084 Salerno, Italy; 5Dipartimento di Scienze della Terra e del Mare, University of Palermo, 90123 Palermo, Italy; 6Food Sciences Laboratory, Formulation Innovation Valorization and Artificial Intelligence (SAFIVIA), Constantine 25000, Algeria

**Keywords:** biodegradable packaging, pectin films, citrus essential oils, antioxidant activity, active food preservation

## Abstract

This study evaluated the physicochemical, morphological, and functional properties of pectin-based films incorporated with lemon essential oil (EO) to assess their potential as biodegradable food packaging materials. The results showed that EO incorporation significantly influenced the film’s characteristics. The control film exhibited a smooth surface, while the EO-containing film had a rougher texture with globular structures and interconnected channels, likely representing dispersed EO droplets and matrix alterations. The mechanical analysis revealed increased elongation at break (20.05 ± 0.784%) for EO-incorporated films, indicating improved flexibility, while tensile strength and Young’s modulus decreased, suggesting reduced stiffness. Film thickness increased slightly with EO (0.097 ± 0.008 mm) compared to the control (0.089 ± 0.001 mm), though the difference was not statistically significant (*p* > 0.05). Moisture content decreased in EO-containing films (28.894%) compared to the control (35.236%), enhancing water resistance. Water solubility increased slightly (16.046 ± 0.003% vs. 15.315 ± 0.040%), while the swelling rate decreased significantly (0.189 ± 0.003 vs. 0.228 ± 0.040; *p* < 0.05), indicating greater structural stability in aqueous environments due to the hydrophobic nature of EO. Transparency tests showed that EO slightly increased film opacity (0.350 ± 0.02 vs. 0.290 ± 0.012), aligning with trends in UV-protective materials. The EO-incorporated films also exhibited moderate antibacterial activity against *Staphylococcus aureus* and *Escherichia coli*. Antifungal tests revealed strong inhibition of *Botrytis cinerea* (100%) and moderate inhibition of *Alternaria alternata* (50%) in EO-containing films. These results demonstrate that EO incorporation improves the functional properties of pectin films, enhancing their flexibility, antimicrobial activity, and environmental stability, making them promising candidates for sustainable food packaging applications. These novel active food packaging materials exhibit strong physical properties and significant potential in maintaining food quality and prolonging shelf life.

## 1. Introduction

The development of active packaging, which integrates bioactive compounds into packaging materials, has gained significant traction in recent years due to its ability to preserve food by preventing microbial contamination and oxidative damage. This innovative approach enhances food safety and shelf life while meeting consumer demand for sustainable, minimally processed, and additive-free products [1,2]. The main role of active packaging is the preservation of food from any microbial contamination and oxidative stress. The use of this packaging by consumers and industries has experienced a real boom in recent years, due to its biological origin, above all its properties, especially with regard to improving food safety and shelf life without causing harmful effects. Because of worries about the harmful impact of synthetic additives on human health, modern consumers prefer foods with little to no artificial additions [3,4]. 

An alternate supply for natural components used in the production of bio-based packaging is waste and byproducts [5]. Given their abundance of functional ingredients (vitamin C, fibers, carotenoids, essential oils, and phenolic compounds) and their many uses in the food, cosmetics, nutraceutical, biofuel, and materials production industries, citrus peels are a highly valuable matrix [6]. Citrus fruits are renowned for their medicinal and therapeutic properties, largely attributed to their rich array of bioactive components. Comprising 75–90% water, 6–9% sugars, and with the remainder consisting of pectin, dietary fiber, minerals, and essential oils, citrus fruits offer a healthy and balanced dietary option. Additionally, they are a significant source of carotenoids and flavonoids [7]. According to FAO STAT [8], global citrus fruit production has significantly increased over the years, rising from approximately 51.48 million tons (MT) in 1975 to 158 MT in 2019. However, the waste generated from citrus fruit processing presents an opportunity to harness valuable bioactive constituents such as phenols, flavonoids, carotenoids, and essential oils. Furthermore, in the context of bioeconomy, this waste could be repurposed for the production of biofuels and other valuable commodities [9].

Pectin, a water-soluble anionic heteropolysaccharide of plant origin, is widely used in the food industry, mainly as a gelling, thickening, and stabilizing agent in fruit products [10]. It is extracted from citrus fruits, apples, currants, black currants, quinces, etc. While pectin can produce packaging films with excellent mechanical properties and barrier functions, there is still a need to enhance the functionalities of pectin-based films by incorporating additional bioactive compounds. This enhancement aims to improve their ability to protect food products and extend their shelf life. The inclusion of active compounds in packaging not only boosts the functional properties of the films, but also contributes to prolonging the freshness and quality of the packaged products [11]. 

Essential oils (EOs) are a good example of natural compounds obtained from a wide variety of plant aromatics that have been applied in the manufacture of food packaging [12]. However, these substances have been applied to low-cost non-biodegradable polymers and biopolymers whose treatment is complex [13].Waste recovery not only reduces the ecological impact by minimizing pollution, but also offers new opportunities allowing economic development; furthermore, the recycling of waste from the agro-processing industry not only minimizes pollution and reduces ecological impact, but also offers new opportunities to enable sustainable economic development in many sectors [14].

However, not all of them are appropriate for use in food packaging applications because of their migration qualities, volatility, and unfavorable taste and/or odor [15]. According to Rehman et al. [16], the primary issue with directly adding EOs to active packaging is the migration of active compounds, which further decreases the efficiency of active packaging during the food’s shelf life. Novel encapsulation techniques play a crucial role in enhancing the stability of bioactive compounds, such as essential oils and plant extracts, while minimizing their reactivity and losses. These encapsulation strategies effectively shield bioactive from external factors that can lead to degradation, thereby preserving their functional properties. By incorporating bioactive compounds into protective matrices, encapsulation not only improves their stability, but also controls their release, enhances solubility, and increases bioavailability [17].

In this context, the present study aims to achieve a double valorization of lemon peels by extracting the essential oil, pectin. It further focuses on developing bioactive pectin-based films incorporating *Citrus limon* essential oil and evaluating their mechanical, antioxidant, and physical properties. This work not only contributes to waste valorization, but also offers a sustainable approach to the production of bioactive packaging materials with potential applications in the food industry. Pectin-based biodegradable film was utilized to extend the shelf life of fresh-cut apples, and its impacts on various quality parameters (firmness, browning index, weight loss, and total soluble solids) was assessed.

## 2. Materials and Methods

### 2.1. Materials

Sodium Sulfate Anhydrous Granular was purchased from Loba Chimie (Sydney, Australia). Glycerol and hydrochloric acid were purchased from Isochem Laboratory, Angamaly, Kochi-683573, India. DPPH and Methanol were obtained from Carlo Erba, Milano, Italy.

### 2.2. Extraction of Essential Oil

The extraction of essential oil from lemon peels was conducted using a Clevenger-type hydro-distillation apparatus, following the method described by the European Pharmacopoeia [18]. Freshly shredded lemon peels (100 g) were combined with 1 L of distilled water and subjected to hydro-distillation for 3 h, or until no further essential oil was produced. The distillation process involved heating the water to produce steam, which carried volatile oil compounds from the lemon peels into the condenser. The essential oil was subsequently separated from the distillate in the Clevenger apparatus. The collected essential oil was dried over anhydrous sodium sulfate to remove any residual moisture, ensuring its stability and purity. The dried oil was then transferred to airtight glass vials, sealed with aluminum foil covers to prevent light exposure, and stored at 0 °C to preserve its chemical integrity until further use.

### 2.3. Pectin Extraction

Pectin was extracted from the residual waste generated during the essential-oil extraction process, using an acidified aqueous method (Figure 1). The extraction was carried out in a hot hydrochloric acid solution under controlled conditions of 45 °C and a pH of 1.25 to hydrolyze and solubilize pectin from the lemon peel matrix. This process facilitates the release of pectin by breaking down the cell wall structure and optimizing its recovery. After the extraction, the pectin was precipitated by the addition of 96% ethanol, which reduced the solubility of pectin in the acidic solution, causing it to aggregate and separate from the liquid phase. The precipitated pectin was collected by filtration, washed with ethanol to remove impurities, and dried.

The yield of pectin was quantified as the percentage of dried pectin obtained relative to the initial mass of lemon peel powder used for the extraction. The yield (Y%) was calculated using the formula(1)$$Y\%\;=\;\frac{M2}{M1}\times 100$$
where Y (%) represents the yield, *M*2 is the mass of the dried pectin obtained (g), and *M*1 is the mass of the dried lemon peel powder introduced into the reactor (g). This method ensures the effective recovery of high-quality pectin from lemon peel residues, highlighting the potential of agro-industrial byproducts for producing valuable biopolymers.

### 2.4. Encapsulation of the Bioactive Compounds in Pectin

Lemon essential oil was encapsulated using pectin as the carrier matrix. Pectin was dissolved to a concentration of 5% (*w*/*v*) in an aqueous solution containing 0.5% (*w*/*v*) citric acid. Lemon essential oil (1%, *v*/*v*) was added to the pectin solution, and the mixture was homogenized using a magnetic stirrer at 4500 rpm for 30 min at 50 °C to ensure uniform dispersion of the essential oil droplets within the pectin matrix. The emulsified samples were frozen at −70 °C for 24 h to stabilize the emulsion structure and prepare it for freeze-drying. Subsequently, the frozen samples were subjected to lyophilization at −55 °C under a vacuum pressure of 0.15 mm Hg for 48 h to remove water content and obtain stable microcapsules in powder form. Control microcapsules without essential oil (EO) were prepared following the same procedure, for comparison. The resulting powdered microcapsules were stored in dark, airtight vials and kept in a freezer until further use to protect them from light and moisture, following the protocol of Jamdar et al. [19].

### 2.5. Preparation of Pectin-Based Biodegradable Films

Pectin-based biodegradable films were prepared using the casting method (Figure 2). Freeze-dried microcapsules were dissolved in an aqueous solution containing 1% (*w*/*v*) citric acid, maintaining a final pectin concentration of 2% (*w*/*v*). The solution was stirred continuously at 50 °C for 30 min, to ensure complete dissolution. To improve the mechanical properties of the films, 5% (*w*/*w*) glycerol was added as a plasticizer, and the mixture was heated at a constant temperature of 45 °C for an additional 15 min. The resulting film-forming solution was degassed using a sonicator to remove entrapped air bubbles, following the method described by Adilah et al. [5]. The degassed solution was then poured into Petri dishes and evenly spread by casting. The films were dried at 40 °C for 24 h to achieve initial solidification. After drying, the films were crosslinked by applying 65 mL of a 2% (*w*/*v*) calcium chloride (CaCl_2_) solution to the surface for enhanced stability. The crosslinked films were subjected to an additional drying step at 40 °C for 30 s to remove excess moisture and facilitate easy removal from the dishes. The films were subsequently stored in a desiccator at room temperature for two days to equilibrate and remove residual moisture before being analyzed, following the protocol of Wang et al. [20].

### 2.6. Properties of Pectin Film

#### 2.6.1. Film Thickness Measurement

The thickness of the pectin-based films was measured using a digital micrometer (ABS ASIMETO, Istanbul, Turkey) with a precision of 0.01 mm. Measurements were taken at nine random points on each film sample, and the mean value was recorded to ensure accuracy and consistency in thickness evaluation.

#### 2.6.2. Moisture Content

The moisture content of the films was determined by measuring the weight loss after drying the samples in an oven at 105 °C until a constant weight was achieved. The moisture content, expressed as a percentage, was calculated using the following equation:(2)$$\mathrm{Moisture}\;\mathrm{content}=\frac{Mw-Md}{Mw}\times 100$$where *Mw* is the weight of the films equilibrated at 75% relative humidity, and *Md* is the dry weight after oven drying. Results were presented as averages of triplicate measurements [21].

#### 2.6.3. Water Solubility and Swelling Rate

For water solubility (*WS*), 2 × 2 cm film samples were pre-dried at 105 °C for 24 h, and the initial dry matter (*W*0) was recorded. The films were submerged in 50 mL of distilled water at either 25 °C or 80 °C in a water bath shaker for 24 h. Undissolved film residues were collected on Whatman filter paper, dried at 105 °C for 24 h, and reweighed (*Wf*). The *WS* percentage was calculated as
(3)$$WS\%=\frac{W0-Wf\;}{W0}\times 100$$*W*0 and *Wf* are the original dry matter and undissolved dry matter, respectively. Alltests were carried out in triplicate. 

For swelling ratio determination, pre-weighed film samples were immersed in water at 25 °C for 2 min. Excess surface water was removed with filter paper, and the samples were reweighed. The swelling ratio was expressed as the percentage weight increase relative to the initial film weight, following the method of Zareen et al. [22].

#### 2.6.4. Film Opacity 

The light barrier properties of the films were assessed using a Cary 60 UV-Vis Spectrophotometer (Agilent, Santa Clara, CA, USA). Rectangular film strips (9 × 40 mm) were analyzed at 600 nm. Film opacity (A) was calculated using the following equation:(4)$$A=\frac{Abs\;(600)}{L}$$
where *Abs*(600) is the absorbance at 600 nm, and *L* is the film thickness in millimeters. 

#### 2.6.5. Biodegradability Test

The biodegradability of the films was evaluated by monitoring weight loss in a composting environment. Film samples (30 × 30 mm) were weighed (*mi*) and buried at a depth of 12–15 cm in compost. Every 5 days, samples were retrieved, rinsed with distilled water, dried at 50 °C for 6 h, and reweighed (*mf*). The percentage mass loss was calculated as
(5)$$\mathrm{Mass}\;\mathrm{loss}\;(\%)=\frac{mi-mf}{mi}\times 100$$

The experiment was conducted over 30 days with quintuplicate measurements, following Wu et al. [23]. 

#### 2.6.6. Measurement of Antioxidant Activity by DPPH

The antioxidant activity of the films was assessed using the DPPH radical scavenging assay. A DPPH solution (120 μM in methanol) was prepared, and 5 μL of the film-forming solution (0.5–2 mg/mL) was mixed with 195 μL of DPPH solution in a 96-well microplate. After incubation at room temperature in the dark for 90 min, absorbance at 515 nm was recorded using a Cary series UV-Vis spectrophotometer. The radical scavenging activity was calculated as
(6)$$\mathrm{Scavenging}\;\mathrm{Activity}=\frac{A_{0\;}-A_{s\;}}{A_{0\;}}\times 100$$
where *A*_0_ is the absorbance of the blank control, and *A_s_* is the absorbance of the sample [24].

#### 2.6.7. Fourier-Transform Infrared (FTIR) Analysis

FTIR spectroscopy was used to evaluate structural interactions between pectin and lemon essential oil in the films. Spectra were recorded using a PerkinElmer FTIR spectrometer, Villepinte, France in the range of 4000–600 cm^−1^ with a resolution of 1 cm^−1^ [25,26]. 

#### 2.6.8. Film Morphology

The surface morphology of the films was analyzed using a VEGA 3 TESCAN, Orsay, France field emission scanning electron microscope (SEM). Prior to imaging, samples were sputter-coated with a thin layer of gold. Imaging was conducted at an accelerating voltage of 20.0 kV under high vacuum.

#### 2.6.9. Mechanical Properties

Mechanical properties, including tensile strength, elongation at break, and Young’s modulus, were determined using a Shimadzu AGS-X universal testing machine, Dardilly, France following ASTM D882 standards [27]. A 1 kN load cell was used, and tests were performed at a crosshead speed of 10 mm/min. Results were calculated as averages from five replicate measurements, with standard deviations reported.

### 2.7. Antibacterial Activity

The antibacterial properties of the films were tested against four bacterial strains: *Staphylococcus aureus* (ATCC 25923), *Bacillus cereus* (ATCC 6633), *Escherichia coli* (ATCC 25922), and *Pseudomonas aeruginosa* (ATCC 2785). Film discs (6 mm diameter) were cut under sterile conditions and placed on Mueller–Hinton agar plates inoculated with 10810^8108 CFU/mL of bacteria. Plates were incubated at 37 °C for 18 h. Zones of inhibition were measured to assess antibacterial activity, with ciprofloxacin serving as the positive control. Tests were performed in quintuplicate for reliability [28].

To assess the antibacterial properties of the pectin film samples, discs with a diameter of 6 mm were cut from the films by using a paper puncher under sterile condition and placed on the surface of Mueller–Hinton agar plates inoculated with 0.1 mL of a bacterial culture containing approximately 108,108 colony-forming units (CFUs)/mL. The plates were incubated at 37 °C for 18 h, and antibacterial activity was determined by measuring the zone of inhibition around each film disc (Table 1). Films without carvacrol served as negative controls. Each inhibition zone test was performed in quintuplicate to ensure the reliability of the results [28]. Ciprofloxacin was used as a positive control.

### 2.8. Antifungal Activity

The antifungal activity of pectin-based films was assessed against two phytopathogenic fungi: *Alternaria alternata* and *Botrytis cinerea*. The inhibitory effects of the films, with and without lemon essential oil (EO), on fungal mycelial growth were evaluated by measuring radial growth on Potato Dextrose Agar (PDA) medium. A 5 mm diameter disk from a young fungal culture was aseptically placed at the center of a Petri dish containing PDA. Film samples were placed on the surface of the fungal inoculum. Dimethyl sulfoxide (DMSO) served as the control. Each treatment was conducted in quintuplicate to ensure statistical reliability [30]. The Petri dishes were incubated at 25 °C for six days, after which the radial growth of fungal colonies was measured in millimeters. The percentage inhibition of mycelial growth for each fungus was calculated relative to the mean colony diameters observed in control plates. The following formula was used: (7)$$I=\frac{C-T}{C}\times 100$$
where:*I*: Inhibition rate (%).*C*: Radial growth of the fungal colony in the control group (DMSO only).*T*: Radial growth of the fungal colony in the presence of the tested film sample.

### 2.9. Application for Preserving Fresh-Cut Apples

#### 2.9.1. Sample Preparation

The method for distributing fresh-cut apples followed the procedure described by Jiang et al. [30], with slight modifications. Fresh apples were thoroughly washed with distilled water, peeled, and cut into pieces measuring 4 × 1 × 1 cm. Film samples were cut into uniform 10 cm × 10 cm strips, and each film was formed into a 3-sided sealed pouch using a heat sealer machine (Runruii plastic film sealer PFS-200, China). Each pouch contained three randomly selected apple slices from a tray holding 40 slices. The samples were divided into two treatment groups: (1) Test Group: apple slices wrapped with pectin film embedded with EO and (2) Unwrapped Group: apple slices without film for comparison. The sealed pouches were stored in a refrigerator at 4 °C with 75% relative humidity (RH), as well as the unwrapped samples. All samples were analyzed at T0 and after a period of 10 days.

#### 2.9.2. Physico-Chemical Traits

Apple weight loss was determined at each sampling date, and expressed as the percentage loss of initial weight. Total soluble solid content (TSS, °Brix) was measured by homogenizing the apple slice and testing the filtrate with a digital refractometer (SinergicaSoluzioni, DBR35, Pescara, Italy). Apple color parameters were measured using a Minolta colorimeter (CR5, Minolta Camera Co., Osaka, Japan) and color changes were reported as lightness (L*) and chroma (C*). A digital penetrometer (TR Snc., Forlì, Italy) equipped with an 8mm probe, was used to test for apple-slice firmness, the penetration depth was set at 5 mm, and the testing speed was 1 mm/s.

### 2.10. Data Analysis

All experiments were conducted in triplicate, and the results are presented as mean ± standard deviation (SD). Statistical analyses were performed using IBM SPSS Statistics v22. Analysis of variance (ANOVA) was used to determine the statistical significance of differences between treatments, with a significance level set at *p* < 0.05. Additionally, the IC_50_ values (concentration required to inhibit 50% of fungal growth) were computed using the Probit model in SPSS.Statistical significance among pectin-coated and uncoated apple slices was analyzed by one-way analysis of variance (ANOVA), and Duncan’s test at 5% level was calculated to compare the differences between means.

## 3. Results and Discussion

### 3.1. Properties of Pectin Film

#### 3.1.1. Film Thickness Measurement

Table 2 presents the thickness of pectin films with and without lemon essential oil (EO). The EO-incorporated films exhibited a thickness of 0.0972 ± 0.008, compared to 0.0894 ± 0.001 for the control films. The difference was not statistically significant (*p* > 0.05). Similar findings were reported by Wu et al. [31] and for gelatin films containing seaweed extract, indicating no significant variations in thickness due to the addition of natural components.

The thickness of a film plays a critical role in determining its mechanical and permeability properties. It is generally influenced by the preparation techniques and drying conditions [32]. Jiang et al. [30] noted that an increase in solid content in the film-forming solution results in thicker films, as observed in this study. Furlan et al. [33] also observed structural and thickness modifications in pectin films incorporated with thyme essential oil, supporting these findings. This result suggests that the addition of EO does not drastically alter the film’s fundamental characteristics, thus maintaining the desired properties for food packaging applications.

#### 3.1.2. Moisture Content

The moisture content of the films was significantly affected by the incorporation of EO (*p* < 0.05), as shown in Table 2. EO-containing films exhibited a moisture content of 28.894%, slightly lower than that of the control films (35.236) This decrease in moisture content is consistent with studies on other biopolymer films, such as those by Wang et al. [34] on chitosan films incorporating tea polyphenols. The reduced moisture content can be attributed to decreased hydrogen bonding between the biopolymer matrix and water molecules when EO is incorporated.

The reduction in moisture content may be beneficial for food packaging applications, as it could improve the film’s shelf life by reducing water activity, which in turn minimizes microbial growth and enzymatic reactions. This result underscores the potential of EO as a functional additive to enhance the moisture resistance and overall durability of pectin-based films.

#### 3.1.3. Water Solubility and Swelling Rate

Water solubility is a critical factor for food packaging applications as it determines the film’s ability to interact with moisture and the environment. Table 2 shows that EO-incorporated films had slightly higher solubility (16.046 ± 0.003% vs. 15.315 ± 0.040% for control films; *p* < 0.05). This is consistent with the study conducted by Venkatachalam et al. [35], which examined the influence of pomelo pericarp essential oil on the structural characteristics of gelatin–arrowroot tuber flour-based edible films. They reported that control films exhibited a solubility efficiency of 41.98%, while gelatin–arrowroot tuber flour films incorporated with pomelo essential oil (PEO) showed solubility levels ranging from 36.56% to 30.86% at PEO concentrations of 0.5% and 2.0%, respectively. This indicates that the addition of PEO effectively reduced the solubility of the films compared to the control, highlighting its potential impact on film properties. In food packaging, water-soluble films may be preferred in applications where biodegradability is a priority, as they can be broken down in the presence of moisture, reducing environmental impact.

The swelling rate was significantly lower in EO-containing films (0.189 ± 0.003) compared to control films (0.228 ± 0.040; *p* < 0.05), likely due to the hydrophobic nature of EO and the structural changes in the film matrix. This reduced swelling rate suggests that the EO acts to stabilize the film matrix, possibly by decreasing the film’s capacity to absorb water. These findings align with a study by Kurtfaki et al. [36] and suggest that EO incorporation enhances the film’s structural stability in aqueous environments.

#### 3.1.4. Film Transparency

Film transparency is another vital property, particularly for food packaging, as it can affect the appearance of packaged products and offer protection against UV radiation. The opacity of films with and without EO was measured and found to be 0.350 ± 0.02 and 0.290 ± 0.12, respectively (Table 2). While the difference was not statistically significant (*p* ≥ 0.05), the trend aligns with previous studies indicating that EO incorporation often increases film opacity. Adilah [5] observed similar opacity levels in biodegradable films containing mango peel extracts, while Norajit et al. [37] reported variations in transparency due to interactions between extracts and alginate films. The results suggest that EO incorporation slightly decreases transparency. The increased opacity may offer UV protection, which is particularly important for products sensitive to light, such as vitamins, oils, or certain fruits and vegetables. 

#### 3.1.5. Biodegradability Test

The biodegradability of pectin films, with and without the incorporation of essential oil (EO), was evaluated over 20 days of incubation (Figure 3). The results indicate that the EO-incorporated films demonstrated a significantly slower degradation rate compared to the control films. After 10 days, surface cracking was observed in both types of films, becoming more pronounced by day 15. By day 20, control films exhibited higher degradation rates, accompanied by smaller pore sizes, compared to EO-incorporated films. This reduced degradation suggests that EO functions as a protective barrier, limiting microbial access and enzymatic activity.

These findings align with the study by He et al. [38], which demonstrated slower degradation in multilayer films containing cinnamon essential oil (SCO) compared to monolayer films. The hydrophobic nature of EO likely reduces water penetration, enhancing the durability of the biofilm. Furthermore, the structural interactions between EO and the pectin matrix strengthen the film and hinder microbial decomposition.

The reduced biodegradation of EO-incorporated films highlights their potential for food packaging applications. By prolonging the durability of packaging films, EO incorporation can help extend food shelf life, minimize food waste, and align with sustainable practices. This approach also valorizes agricultural byproducts, such as lemon peels, contributing to a circular economy.

#### 3.1.6. Test of DPPH

The DPPH scavenging activity of pectin films is presented in Table 3. Films containing EO exhibited significantly higher antioxidant activity (63.60 ± 0.001%) compared to control films (21.37 ± 0.001; *p* < 0.05). These results are consistent with Kumar et al. [39], who observed enhanced antioxidant capacity in chitosan films incorporated with essential oils. The increased scavenging activity can be attributed to the natural antioxidant properties of lemon EO, which contains bioactive compounds such as limonene and geraniol. Reducing biomolecules to the nanoscale further amplifies their surface area and reactivity, as highlighted by Rehman et al. [16]. This enhancement in antioxidant activity underscores the effectiveness of EO as a functional component in active food packaging systems.

#### 3.1.7. FTIR Analysis

Fourier-transform infrared (FTIR) spectroscopy was performed to analyze the chemical interactions in the pectin films, with and without EO, over a wavenumber range of 600–4000 cm^−1^ (Figure 4). In the control film, prominent peaks were observed between 3000 and 3500 cm^−1^, corresponding to O-H stretching vibrations, indicative of moisture content. The peak at 2925 cm^−1^ represents CH₂ stretching vibrations in polysaccharides, while the peak at 1734 cm^−1^ is associated with C=O vibrations from carboxylic acids such as glucuronic acid. Other notable peaks included those at 1603 cm^−1^ (C=C aromatic vibrations), 1437 cm^−1^ (C-H bending), and 1328 cm^−1^ (C-O stretching). The FTIR spectrum of the EO-incorporated film exhibited a shift in the carbonyl peak to 1715 cm^−1^, characteristic of C=C vibrations from compounds like limonene, citronellol, and geraniol present in lemon EO. This shift confirms the successful incorporation of EO into the pectin matrix. The spectral differences observed between the control and EO-containing films highlight the chemical interactions between pectin and EO, which enhance the film’s functional properties, including its antioxidant and antimicrobial activities.

#### 3.1.8. Film Morphology

The surface morphology of pectin films with and without essential oil (EO) was analyzed using SEM micrographs (Figure 5). The control film exhibited a smooth surface with minor variations, indicating a network of fine, thread-like structures that likely represent the pectin biopolymer. This smooth and uniform appearance is characteristic of pure pectin films, suggesting a cohesive polymer network. In contrast, films incorporated with EO displayed a rougher surface texture, characterized by globular structures that are likely EO droplets dispersed within the pectin matrix. Furthermore, interconnected channels were observed throughout the film, which could have resulted from the interaction and incorporation of EO into the biopolymer. Both film types exhibited a grainy structure, potentially caused by molecular changes in the pectin during film formation. These findings are consistent with previous studies. Nisar et al. [40] reported that the incorporation of chitosan nanoparticles into pectin films led to significant morphological changes, including a more complex microstructure. Similarly, Furlan et al. [33] observed that the addition of thyme essential oil to pectin films altered the surface morphology, leading to increased roughness, due to interactions between the EO and the polymer chains. These morphological differences highlight the impact of EO addition on the structural characteristics of pectin films. The changes observed could influence the functional properties of the films, such as barrier performance and mechanical behavior, which are crucial for their application in food packaging. 

#### 3.1.9. Mechanical Properties

The mechanical properties of the films revealed notable differences between the control film and the film with essential oil (EO) (Table 4). The control film exhibited higher tensile strength (12.28 ± 1.271 MPa), indicating a stronger, more rigid structure. In contrast, the film with EO demonstrated a slightly higher elongation at break (20.05 ± 0.784%) and a lower Young’s modulus (174.73 ± 1.915 MPa), suggesting enhanced flexibility and reduced stiffness. These changes are likely due to the incorporation of EO, which may act as a plasticizer, reducing polymer chain interactions within the pectin matrix.

The tensile strength of the film with EO was comparable to findings by Jiang et al. [30], who reported tensile strength values of 31.26 ± 2.30 MPa for lemon peel pectin films. Additionally, the observed elongation aligns with studies on mandarin peel pectin films, where elongation values reached 17.26 ± 1.67% [41]. The results are consistent with Baghi et al. [42], who reported a decrease in tensile strength and Young’s modulus after incorporating nanoemulsions into pectin films, with a corresponding increase in elongation at break. However, the observed trends deviate from Vahedikia et al. [43], who found that incorporating cinnamon essential oil with chitosan nanoparticles (CEO-CNPs) increased tensile strength but reduced elongation in zein film composites. These differences may be attributed to the nature of the essential oils, film composition, and interaction between EO and the polymer matrix.

The mechanical properties of the EO-incorporated films suggest their suitability for flexible packaging applications. However, their reduced stiffness may limit their use in applications requiring high mechanical strength.

### 3.2. Antibacterial Activity

The antibacterial efficacy of the films was tested against Gram-positive bacteria (*Staphylococcus aureus and Bacillus cereus*) and Gram-negative bacteria (*Escherichia coli and Pseudomonas aeruginosa*) (Table 5, Figure 6). The control film exhibited no antibacterial activity, with zero zones of inhibition for all bacterial strains. Conversely, the film with EO showed moderate antibacterial activity, with a zone of inhibition of 0.930 ± 0.574 mm against Staphylococcus aureus and 0.750 ± 0.203 mm against *Escherichia coli*. However, no inhibition was observed for *Bacillus cereus* or *Pseudomonas aeruginosa*, suggesting that the efficacy of the EO varies, depending on the bacterial strain. These results align with studies highlighting the variability in antibacterial effectiveness of essential oils due to their chemical composition. Gram-negative bacteria, such as E. coli and *Pseudomonas aeruginosa*, are more resistant to EO due to the presence of hydrophobic lipopolysaccharides in their outer membrane, which limit the diffusion of hydrophobic compounds [44,45]. In contrast, Gram-positive bacteria like S. aureus lack this outer membrane, allowing essential oils to penetrate more effectively through their peptidoglycan cell wall. These findings are consistent with Bharti et al. [46], who reported that starch bio-based composite edible films functionalized with Carum carvi L. essential oil showed greater antibacterial activity against Gram-positive bacteria compared to Gram-negative bacteria. The enhanced antibacterial properties of the EO-incorporated films indicate their potential for use in food packaging, to enhance safety and extend shelf life.

### 3.3. Antifungal Activity

The antifungal activity of the pectin films was assessed against *Alternaria alternata* and *Botrytis cinerea*, as shown in Figure 7 and summarized in Table 6. The antifungal properties of these films are critical for their potential use in food packaging, where they can enhance the safety and preservation of food products. The results showed that the films with essential oils (EOs) exhibited stronger inhibition against *Botrytis cinerea*, with 100% inhibition for the film containing lemon EO compared to only 70% inhibition by the control film. On the other hand, when tested against *Alternaria alternata*, the control film showed a modest 10% inhibition, while the film with EO demonstrated a 50% inhibition rate. These results suggest that pectin films, particularly those incorporated with EO, possess inherent antifungal properties, especially against *Botrytis cinerea*.

These findings align with previous studies, such as that by Alvarez et al. [47], which investigated natural pectin-based edible coatings with antifungal properties aimed at controlling green mold in ‘Valencia’ oranges. Their study showed that these coatings successfully reduced postharvest losses and maintained fruit quality by inhibiting fungal growth, highlighting the potential of pectin films as an effective and sustainable approach for managing postharvest diseases in citrus fruits.

### 3.4. Effect of Pectin–EO-Based Film on Physico-Chemical Features

Cumulative weight loss showed higher values in the uncoated apples compared to our film-coated apple, with the progression of the storage period. After 10 days of cold storage, the weight loss of fresh-cut apples increased significantly over the storage period, and exhibited a weight loss of about 38.5% ± 1.25 after 10 days, primarily due to respiration and stomatal transpiration. Apples packaged in pectin–EO film showed a lower weight loss (5.2% ± 0.81), while pectin-based films provided better oxygen barrier properties, reducing respiration intensity and water loss.

Firmness typically decreases over time, as the apple slices lose moisture and undergo cellular changes. The loss of water and the breakdown of cell walls due to respiration and enzymatic activity contribute to a softening of the fruit. At the end of cold storage, pectin-coated apple slices retained significantly higher firmness compared to untreated ones; after 10 days, apples which were uncoated showed a 57.5% ± 2.04 reduction in firmness, whereas apples in pectin–EO-based films experienced slower firmness loss (12.1% ± 1.94).

The TSS loss in apples packaged in pectin–EO film was slower than those which were uncoated, suggesting that pectin coatings reduce the rate of respiration by acting as a barrier to oxygen and moisture loss. This helps to slow down the consumption of sugars and other soluble solids. Pectin-coated slices experienced a 12.04% ± 0.93 increase in TSS during 10 days of cold storage, while uncoated slices showed a 24.46% ± 2.12 increase during the same periods.

The results also demonstrated significant effects of both storage conditions and pectin coating on color parameters, such as lightness (L*) and chroma (C). On the 10th day, the BI of the wrapped group was the lowest, indicating that pectin–EO film effectively inhibited the browning of the apples. Over 10 days of cold storage, uncoated apple slices showed a decline in chroma (C) of 19.1% ± 1.10 and in L value* of 21.2% ± 2.21, as the fruit loses its color intensity due to the effects of moisture loss and enzymatic activity. Pectin-coated apple slices retained a higher L value* (6.2% ± 0.94) and showed less decrease in chroma (C) (5.1% ± 0.91). This could be attributed to the antioxidant properties of the lemon essential oil, which reduces polyphenol oxidase activity, a major cause of browning in fresh-cut apples. Additionally, the lower oxygen permeability of pectin film further helped reduce enzymatic browning by limiting oxygen exposure. This makes the pectin–EO film a useful coating material for preserving the aesthetic quality of fruit during cold storage.

## 4. Conclusions

This study demonstrates that *citrus limon* essential oil can be effectively encapsulated and incorporated into biodegradable pectin films, enhancing their antioxidant activity, particularly in free-radical scavenging. The addition of lemon essential oil did not significantly affect the water solubility, transparency, or thickness of the films, though differences in moisture content and a reduction in biodegradability were observed compared to the control film. SEM analysis revealed a rougher texture with distinct globular structures in the EO-incorporated films, in contrast to the smoother surface of the control film. FT-IR spectroscopy confirmed that both films maintained the same chemical structure, with no changes in functional groups. Mechanical testing indicated that the EO-encapsulated films exhibited increased tensile strength and elongation at break, suggesting enhanced flexibility, although their Young’s modulus was lower than that of the control film. Additionally, these films demonstrated antibacterial activity against *Staphylococcus aureus* and *Bacillus cereus*, as well as antifungal activity against *Alternaria alternata* and *Botrytis cinerea*. These findings underscore the potential of citrus limon essential-oil-enriched pectin films as active packaging solutions for food applications, warranting further investigation into their practical efficacy in real-world scenarios. While this study provides promising results regarding the incorporation of citrus limon essential oil into pectin films, several limitations should be considered. First, the study primarily focused on the laboratory-scale preparation of the films, and the results may vary when scaled up for commercial production. Furthermore, the films’ mechanical properties were measured under standard conditions, but they could behave differently under dynamic or mechanical stress encountered in practical applications. The pectin film proved to be an effective coating for extending the shelf life of apple slices by significantly reducing moisture loss, slowing enzymatic browning, and preserving key quality attributes such as firmness, TSS, and color, thereby maintaining the fruit’s freshness and appearance during storage.

## Figures and Tables

**Figure 1 foods-14-00353-f001:**
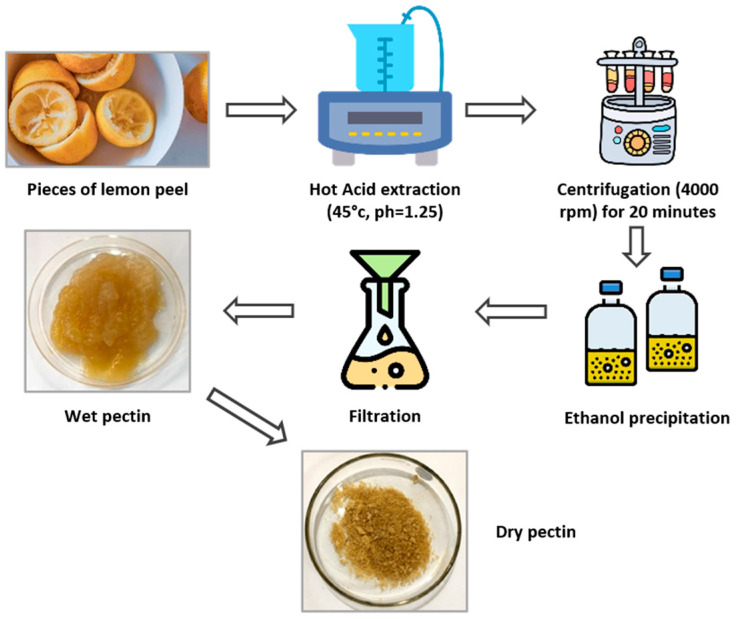
Schematic design for pectin extraction.

**Figure 2 foods-14-00353-f002:**
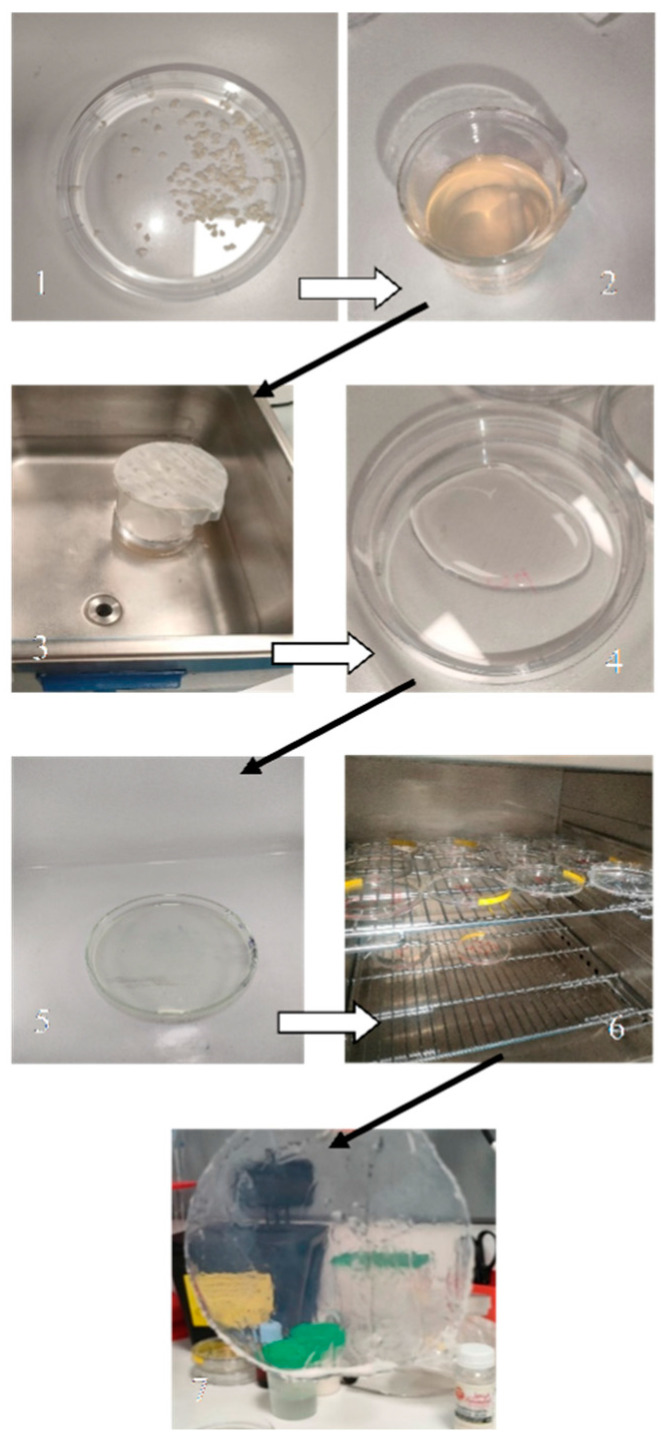
Preparation of pectin-based biodegradable films. 1—encapsulation of the essential oil in pectin; 2—preparation of pectin based biodegradable films; 3—degassing of the film-forming solutions remove air bubbles using a sonicator; 4–5 distribution of the film solution into Petri dishes for casting; 6—dried pectin film; 7—pectin-based biodegradable films.

**Figure 3 foods-14-00353-f003:**
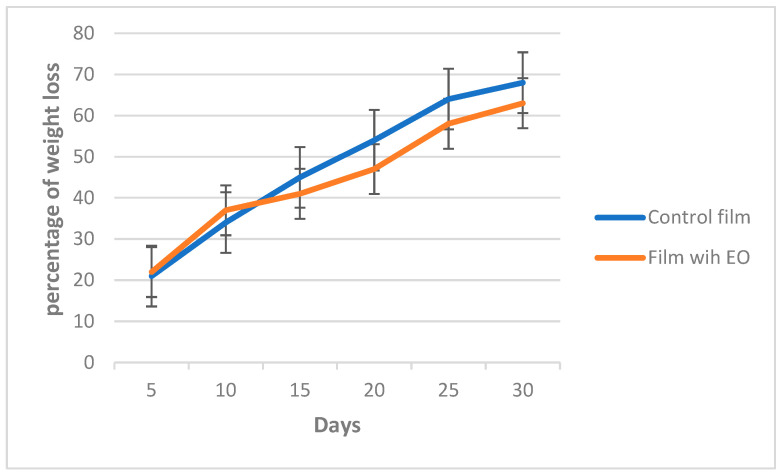
Weight loss versus time of film with EO and control film.

**Figure 4 foods-14-00353-f004:**
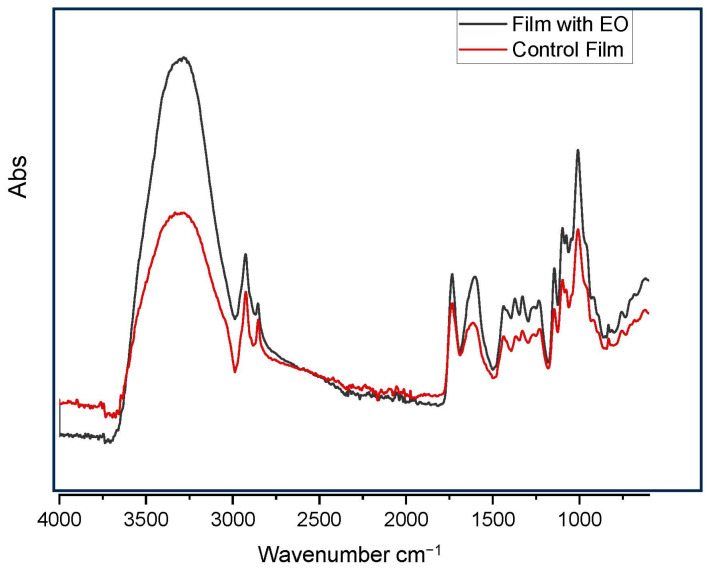
FTIR spectroscopy of pectin film.

**Figure 5 foods-14-00353-f005:**
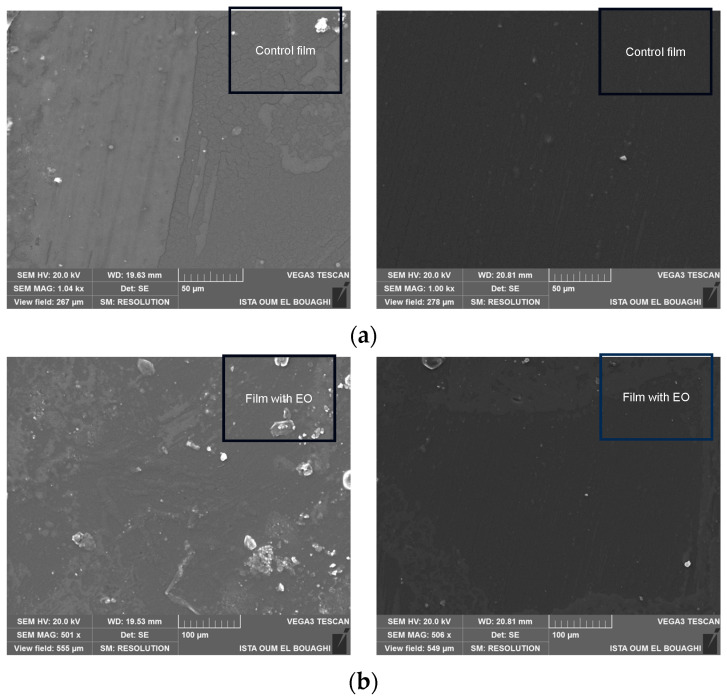
SEM micrographs of surface (**a**): control film; (**b**): film incorporated with EO.

**Figure 6 foods-14-00353-f006:**
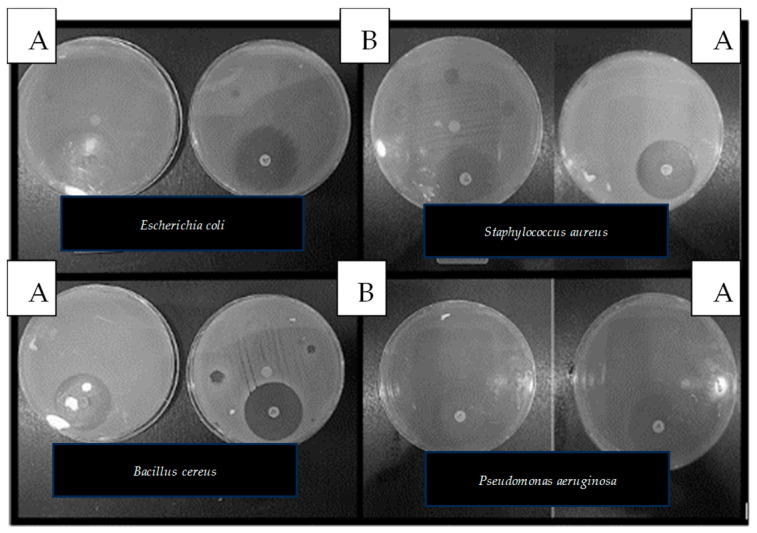
Antibacterial activity of the pectin films: A is the control film, B is the film with EO.

**Figure 7 foods-14-00353-f007:**
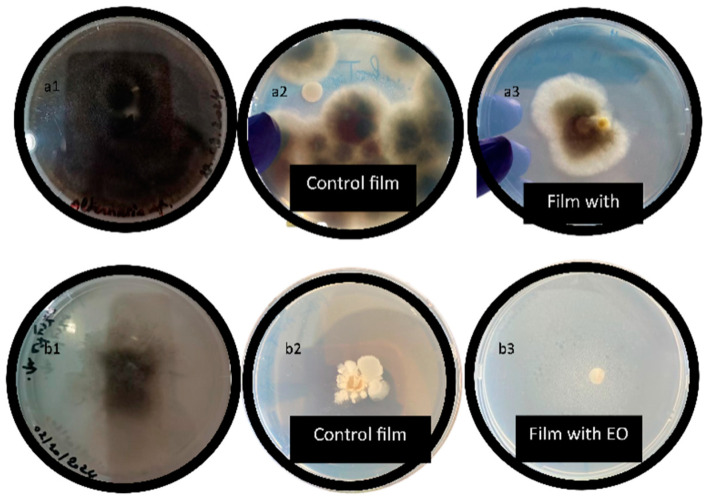
Antifungal activity of pectin films (**a1**) *Alternaria alternata* with DMSO, (**a2**) *Alternaria alternata* control film, (**a3**) *Alternaria alternata* film with EO, (**b1**) *Botrytis cinerea* with DMSO, (**b2**) *Botrytis cinerea* control film, (**b3**) *Botrytis cinerea* film with EO.

**Table 1 foods-14-00353-t001:** Scale for assessing antibacterial activity of the films [29].

Characteristics of Strain Growth	Effect
Growth inhibition zone > 1 mm around the film and under the film	+++ Very good activity
Growth inhibition zone < 1 mm around the film or only under the film	++ Good activity
No inhibition zone around the film and slight inhibition under the film	+ Low activity
No growth inhibition around or under the film	− Lack of activity

**Table 2 foods-14-00353-t002:** Properties of Pectin Films (Mean ± SD).

Film Samples	Thickness (mm)	Moisture Content%	Water Solubility%	Swelling Rate	Film Opacity
Control	0.089 ± 0.001	35.236	15.315	0.228 ± 0.040	0.290 ± 0.012
Film with EO	0.097 ± 0.008	28.894	16.046	0.189 ± 0.003	0.350 ± 0.020

**Table 3 foods-14-00353-t003:** DPPH free-radical scavenging activity of pectin films (Mean ± SD).

Film with EO	Film without EO
63.60% ± 0.001	21.37% ± 0.001

**Table 4 foods-14-00353-t004:** Mechanical properties of pectin films (Mean ± SD).

Film Sample	Elongation (%)	Strength (MPa)	Young’s Modulus (MPa)
Control Film	11.52 ± 0.123	12.28 ± 1.271	494.77 ± 0.519
Film with EO	20.05 ± 0.784	14.68 ± 1.419	174.73 ± 1.915

**Table 5 foods-14-00353-t005:** Antibacterial activity of the films (Mean ± SD).

Film	Control Film	Film with EO
*Staphylococcus aureus* (mm)	-	0.930 ± 0.574 (++)
	-	-
*Escherichia coli* (mm)	-	-
	-	-
*Bacillus cereus* (mm)	-	0.750 ± 0.203 (++)
*Pseudomonas aeruginosa* (mm)	-	-
	-	-

++ = Moderate antibacterial activity. - = No inhibition.

**Table 6 foods-14-00353-t006:** Antifungal activity of pectin films (Mean ± SD).

Film	Control Film	Film with EO
Inhibition (%) *Botrytis cinerea*	70 ± 2.500	100 ± 0.000
Inhibition (%) *Alternaria alternata*	10 ± 4.800	50 ± 1.700

## Data Availability

The original contributions presented in the study are included in the article, further inquiries can be directed to the corresponding author.

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
