# Peer review of "Development of Pectin-Based Films with Encapsulated Lemon Essential Oil for Active Food Packaging: Improved Antioxidant Activity and Biodegradation"

_foods, 2025, doi:10.3390/foods14030353_

Round 1
Reviewer 1 Report
Comments and Suggestions for Authors
The project has potential. The pectin films obtained and studied gave promising preliminary results. There are several issues needing immediate attention from the authors.
1. It is unclear how the films incorporating essential oil were prepared. What I understood is the fact that the pectin obtained as a residue (soluble in water) from the extraction of the essential oil from the lemon peels was mixed with citric acid, under stirring, at 45C, then freeze-dried to form capsules. How was the essential oil included? At what concentration?
2. What does RH stay for?
3. What is the measuring unit used for film transparency? The manuscript mentions A/mm. I do not think that it is correct.
4. Equation 5, for mass loss, should be revised.
5. Antioxidant capacity was evaluated with two different free radicals, using 2 different model compounds: gallic acid for DPPH method and Trolox for ABTS method. Experimental details are not clear: a given volume of pectin film was combined with the free radical solution and mixture absorbance was measured before and after a set reaction time, to calculate the inhibition percentage, or the scavenging activity in %. How was the volume of pectin film measured? What does the 0.5 – 2 mg/L concentration range refer to?
6. Information about the quality of antioxidant capacity measurements should be provided in terms of parameters of the calibration curves used, repeatability, accuracy, and recovery for both free radical used (DPPH or ABTS).
7. Results reporting antioxidant capacity should include behavior of samples compared to the model compound, generally Trolox. This is generally expressed as equivalent concentration of model compound, using either micro g/mL, micro M, or micro g/g of sample. IC50 is very often used. The values is not reported in the manuscript.
8. The following fragment needs revision: ‘Trolox Equivalents (TE) measured in millimoles per liter of juice were used to quantify the results.’
There is no juice evaluated in the current project, just the antioxidant capacity of the pectin films. The results should be reported within the context of the present project.
9. Attention when reporting results and corresponding standard errors. They should have the same number of digits. Measuring units should always accompany experimental results.
10. The fragment needs correction: R224 ‘The solubility of the materials is equal for both films15.315% for the one without essential oil and 16.046 for the one with it.’
There are two different experimental values, so solubility in water can not be the same. They are of similar order of magnitude, but they are not equal.
11. The experimental section gives the formula for calculating film opacity. How is it related to transparency? What measuring units can be used?
12. How many repeated measurements were carried out for the biodegradability evaluation?
13. Figure 2 should have a clear Ox axis. The design used is confusing.
14. The authors declared that the pores were smaller in the control film after 20 days. Did they measure the dimensions of the pores?
15. R250-251. The fragment needs reformulation: ‘However, after 20 days, the pores were smaller in the control film, and the degradation increased compared to the film incorporated with EO.’
16. R270: Figure 3a, page 9, contains an IR spectrum, with no connection to the DPPH antioxidant capacity.
17. R270-272: ‘The integrated film was shown to have a considerably stronger scavenging action (p <0.05) against DPPH radicals, with a value that was lower than that of Trolox.”
Since Trolox was the model compound only for ABTS method, the comment is pointless.
18. Results in the DPPH method are generally reported as equivalent of model compound mass per unit of volume. The Inhibition % or Scavenging % is just an intermediate stage in the data processing.
19. Figures related to the comments in the section ‚Test of DPPH’ are missing.
20. There is general knowledge that is not connected to the project reported. This is an example of correct general knowledge, unrelated to the results reported (R267-269): ‚Antioxidants are known to reduce lipid oxidation through a mechanism known as free radical scavenging. Antioxidants have the ability to suppress free radicals, as demonstrated by the DPPH free radical scavenging experiment.’
21. Since model compounds in the two antioxidant capacity assessing methods differ, comparison is not relevant.
22. R273-274: the sentence is confusing ‚The three displayed figures of 21.37% and 63.60%, in that order.’
23. PWR stays for pectin washable residues? It should be explained at the time of the first mention.
24. It is not recommended to use ‚DPPH° neutralizing capacities’ when it comes to the interaction between the free radical and the antioxidants present in the investigated sample.
25. R293-294. Please reformulate the sentence: ‚Pectin washing residues showed a reduction in power activity, which obviously rose as the PWR concentration it can be valorized and used in food's conservation.’
26. R299-300. The statement is unfounded ‚ It is clear that our compound is of high purity, we can observe the presence of the characteristic vibrations of the O-H peak of free water.’
It is not clear what compound the authors are referring to. I could guess that it can be the essential oil. Since this is a mixture of at least 3 chemical compounds (limonene, citronellol, and geraniol, as the authors declare in R316) how can purity be declared based on the presence of the OH vibration bands?
27. Figure 3 on page 9 should have a more elaborate caption, to explain the a and b graphs content.
28. R328: figure 3 is not the current figure number. Rechecking is necessary.
29. There are many editing issues to be addressed: linked words, no space after commas or full stop points.
30. References should be cited according to the journal’s recommendations.
31. Chemical formulas should be written with the corresponding subscripts.
The attached file is a highlighted version of the manuscript, for speedy revision.

Style needs improvement.
Author Response
The project has potential. The pectin films obtained and studied gave promising preliminary results. There are several issues needing immediate attention from the authors.
- It is unclear how the films incorporating essential oil were prepared. What I understood is the fact that the pectin obtained as a residue (soluble in water) from the extraction of the essential oil from the lemon peels was mixed with citric acid, under stirring, at 45°C, then freeze-dried to form capsules. How was the essential oil included? At what concentration?
It was included with the pectin and citric acid at concentration of 1%. Line 150-151
- What does RH stay for?
RH IS Relative Humidity. Line 183
- What is the measuring unit used for film transparency? The manuscript mentions A/mm. I do not think that it is correct.
Film opacity
The light barrier properties of pectin films against visible and ultraviolet (UV) radiation were evaluated by measuring the opacity of the films. A rectangular film strip measuring 9 mm × 40 mm was placed on a Cary series UV-VIS spectrophotometer, and absorbance was recorded at a wavelength of 600 nm. The opacity of the film was calculated using the equation:
A= (5)
where A is the absorbance, Abs is the absorbance at 600 nm and L is the thickness of the film sample (mm). line 200-204
- Equation 5, for mass loss, should be revised.
The biodegradability of the samples was assessed by tracking mass loss over time in a compost environment. Samples measuring 30 mm by 30 mm were weighed and buried in compost boxes at a depth of 12 to 15 cm. The buried samples were removed every five days, cleaned with distilled water, and allowed to dry for six hours at 50°C in an oven. After drying, the samples were reweighed and returned to the compost for a total duration of 30 days. The mass loss was calculated using the following formula:
Mass loss (%) =
Where is the final mass of the tested sample and is the initial mass of the sample tested.
This ratio is generally negative due to weight loss (Wu., 2012). Line 211
- Antioxidant capacity was evaluated with two different free radicals, using 2 different model compounds: gallic acid for DPPH method and Trolox for ABTS method. Experimental details are not clear: a given volume of pectin film was combined with the free radical solution and mixture absorbance was measured before and after a set reaction time, to calculate the inhibition percentage, or the scavenging activity in %. How was the volume of pectin film measured? What does the 0.5 – 2 mg/L concentration range refer to?
- Information about the quality of antioxidant capacity measurements should be provided in terms of parameters of the calibration curves used, repeatability, accuracy, and recovery for both free radical used (DPPH or ABTS).
- Results reporting antioxidant capacity should include behavior of samples compared to the model compound, generally Trolox. This is generally expressed as equivalent concentration of model compound, using either micro g/mL, micro M, or micro g/g of sample. IC50is very often used. The values is not reported in the manuscript.
- The following fragment needs revision: ‘Trolox Equivalents (TE) measured in millimoles per liter of juice were used to quantify the results.’
There is no juice evaluated in the current project, just the antioxidant capacity of the pectin films. The results should be reported within the context of the present project.
5-6-7-8
The authors eliminate the of test of ABTS, furthermore here is corrections of the test of DPPH:
The antioxidant activity of pectin films was evaluated using the DPPH (1,1-diphenyl-2-picrylhydrazyl) radical scavenging assay, following a modified method based on Brand-Williams et al. (1995). The experimental setup involved preparing a 120 μM DPPH solution in methanol.
At room temperature and with protection from light, 5 μL of film forming solution and pectin washing residues with concentrations ranging from 0.5 to 2 mg/mL, was combined with 195 μL of DPPH standard solution at a concentration of 120 μM in a 96-well microplate (Campone et al.,2018). The absorbance of the mixture was measured at 515 nm using a Cary series UV-VIS spectrophotometer after an incubation period of 90 minutes at room temperature, protected from light. The percentage of DPPH radical scavenging activity was calculated using the following formula:
Scavenging Activity = (6)
Where is the absorbance of the sample and is the absorbance of the blank control. Line 215-223.
- Attention when reporting results and corresponding standard errors. They should have the same number of digits. Measuring units should always accompany experimental results.
Table 1. Pectin film proprieties, SD means standard deviation. Line 248
|
Film samples |
Thickness(mm) |
Moisture content% |
Water solubility% |
swelling rate |
Film Opacity |
|
Control |
0.089±0.001 |
35.236 |
15.315 |
0.228±0.040 |
0.290±0.012 |
|
Film with EO |
0.097±0.008 |
28.894 |
16.046 |
0.189±0.003 |
0.350±0.02 |
- The fragment needs correction: R224 ‘The solubility of the materials is equal for both films15.315% for the one without essential oil and 16.046 for the one with it.’
There are two different experimental values, so solubility in water cannot be the same. They are of similar order of magnitude, but they are not equal.
Water solubility is a critical factor for food packaging applications as it determines the film’s ability to interact with moisture and the environment. Table 2 shows that EO-incorporated films had slightly higher solubility (16.046±0.003% vs. 15.315±0.040% for control films; p<0.05). This is consistent with the study conducted by Venkatachalam et al. (35), which examined the influence of pomelo pericarp essential oil on the structural characteristics of gelatin-arrowroot tuber flour-based edible films. They reported that control films exhibited a solubility efficiency of 41.98%, while gelatin-arrowroot tuber flour films incorporated with pomelo essential oil (PEO) showed solubility levels ranging from 36.56% to 30.86% at PEO concentrations of 0.5% and 2.0%, respectively. This indicates that the addition of PEO effectively reduced the solubility of the films compared to the control, highlighting its potential impact on film properties. In food packaging, water-soluble films may be preferred in applications where biodegradability is a priority, as they can be broken down in the presence of moisture, reducing environmental impact. Line 319-329.
- The experimental section gives the formula for calculating film opacity. How is it related to transparency? What measuring units can be used?
Opacity is a critical property in evaluating the performance of films, particularly in food packaging applications. It refers to a material's ability to block the passage of light, which is essential for preventing light-induced degradation of food products. The opacity of a film can be quantified using the formula:
Opacity=100%−Transmittance
Where transmittance is the percentage of light that passes through the film. A higher opacity value indicates that less light is transmitted, making the film more effective at blocking light. The measuring units for opacity are typically expressed as percentage points, where a perfectly opaque material would have an opacity value of 100%
- How many repeated measurements were carried out for the biodegradability evaluation?
Five measurements were carried out for the biodegradability evaluation. Line 212
- Figure 2 should have a clear Ox axis. The design used is confusing.
Figure 2. Weight loss versus time of film with EO and control film.
- The authors declared that the pores were smaller in the control film after 20 days. Did they measure the dimensions of the pores?
In response to the query regarding the measurement of pore dimensions in the control film after 20 days, we would like to clarify that the evaluation of pore size was conducted visually. We assessed the changes in pore dimensions using a qualitative approach, observing the films under standard lighting conditions. While this method provides a general indication of pore size variation, we acknowledge that more precise measurements could be obtained through quantitative techniques such as scanning electron microscopy (SEM) in future studies.
- R250-251. The fragment needs reformulation: ‘However, after 20 days, the pores were smaller in the control film, and the degradation increased compared to the film incorporated with EO.’
The biodegradability of pectin films, with and without the incorporation of essential oil (EO), was evaluated over 20 days of incubation (Figure 3). The results indicate that the EO-incorporated films demonstrated a significantly slower degradation rate compared to the control films. After 10 days, surface cracking was observed in both types of films, becoming more pronounced by day 15. By day 20, control films exhibited higher degradation rates, accompanied by smaller pore sizes, compared to EO-incorporated films. This reduced degradation suggests that EO functions as a protective barrier, limiting microbial access and enzymatic activity.
These findings align with the study by He et al. (38), which demonstrated slower degradation in multilayer films containing cinnamon essential oil (SCO) compared to monolayer films. The hydrophobic nature of EO likely reduces water penetration, enhancing the durability of the biofilm. Furthermore, the structural interactions between EO and the pectin matrix strengthen the film and hinder microbial decomposition.
The reduced biodegradation of EO-incorporated films highlights their potential for food packaging applications. By prolonging the durability of packaging films, EO incorporation can help extend food shelf life, minimize food waste, and align with sustainable practices. This approach also valorizes agricultural byproducts, such as lemon peels, contributing to a circular economy. Line 350-367
- R270: Figure 3a, page 9, contains an IR spectrum, with no connection to the DPPH antioxidant capacity.
We appreciate your observation regarding the figure number. We will recheck .
- R270-272: ‘The integrated film was shown to have a considerably stronger scavenging action (p <0.05) against DPPH radicals, with a value that was lower than that of Trolox.”
Since Trolox was the model compound only for ABTS method, the comment is pointless.
We would like to clarify that Trolox was indeed utilized as a standard in our DPPH radical scavenging assay, not just in the ABTS method.
- Results in the DPPH method are generally reported as equivalent of model compound mass per unit of volume. The Inhibition % or Scavenging % is just an intermediate stage in the data processing.
In response to the question regarding the reporting of results in the DPPH method, we would like to clarify that the primary objective of our study is to evaluate the activity of the pectin films for potential food preservation applications. While results are typically expressed as the equivalent concentration of a model compound, we focused on the percentage of inhibition or scavenging activity as a straightforward indicator of whether the films possess antioxidant activity. This approach allows us to determine if the films can effectively scavenge free radicals, which is crucial for their role in food preservation. Our aim is to formulate active films that enhance food safety and shelf life, and thus, assessing the percentage of scavenging activity serves as a practical measure of their efficacy in this context.
- Figures related to the comments in the section ‚Test of DPPH’ are missing.
In response to the comment regarding missing figures related to the "Test of DPPH," we would like to clarify that we have included a table that summarizes the relevant data. To enhance clarity, we have revised the sentence in the manuscript.
- There is general knowledge that is not connected to the project reported. This is an example of correct general knowledge, unrelated to the results reported (R267-269): ‚Antioxidants are known to reduce lipid oxidation through a mechanism known as free radical scavenging. Antioxidants have the ability to suppress free radicals, as demonstrated by the DPPH free radical scavenging experiment.’
The statements about antioxidants and their mechanisms of action, including lipid oxidation reduction and free radical scavenging, were included to provide context for the significance of our findings. While they may not directly relate to the specific results of our study, they serve to highlight the underlying principles that support the relevance of antioxidant activity in food preservation applications. Therefore , this is the reformulation.
When dissolved in methanol, DPPH (1,1-diphenyl-2-picrylhydrazyl), a stable free radical, exhibits a distinctive absorption peak at 517 nm, which gives the solution its characteristic purple color. As antioxidant molecules scavenge the DPPH radicals through hydrogen donation, the absorbance decreases, resulting in a color change from purple to light yellow. This mechanism of free radical scavenging is well-documented as a means by which antioxidants can reduce lipid oxidation.
- Since model compounds in the two-antioxidant capacity assessing methods differ, comparison is not relevant.
We would like to clarify that we have eliminated the ABTS method from our study. By focusing solely on the DPPH method, we aim to provide a more consistent and relevant assessment of antioxidant activity using a single model compound.
- R273-274: the sentence is confusing ‚The three displayed figures of 21.37% and 63.60%, in that order.’
Films containing EO exhibited significantly higher antioxidant activity (63.60±0.001%) compared to control films (21.37±0.001; p<0.05p). Line 372-373
- PWR stays for pectin washable residues? It should be explained at the time of the first mention.
We eliminate this part.
- It is not recommended to use ‚DPPH° neutralizing capacities’ when it comes to the interaction between the free radical and the antioxidants present in the investigated sample.
In response to the comment regarding the use of "DPPH° neutralizing capacities" in our study, we would like to clarify that our focus is on assessing the antioxidant activity of the samples through their ability to scavenge free radicals. While we understand that the term "neutralizing capacities" may not be the most appropriate description for this interaction, our intention was to convey how effectively the antioxidants present in the samples can reduce DPPH radicals. The DPPH assay is a well-established method for evaluating antioxidant activity, and it specifically measures how antioxidants can donate hydrogen atoms to stabilize free radicals, leading to a decrease in absorbance at 517 nm. We appreciate your feedback and will consider revising our terminology to ensure it accurately reflects the nature of these interactions in future revisions.
- R293-294. Please reformulate the sentence: ‚Pectin washing residues showed a reduction in power activity, which obviously rose as the PWR concentration it can be valorized and used in food's conservation.’
We eliminate this part.
- R299-300. The statement is unfounded ‚ It is clear that our compound is of high purity, we can observe the presence of the characteristic vibrations of the O-H peak of free water.
We would like to clarify that our assertion was made in the context of identifying specific functional groups and molecular interactions observed in the spectra. While we noted the presence of the O-H peak associated with free water, which can indicate moisture content, we understand that this alone does not conclusively determine the overall purity of the compound. The presence of characteristic peaks in FTIR analysis does suggest specific functional groups related to the composition of the pectin film, but further analytical techniques may be necessary to comprehensively assess purity.We have reformulate the discussion.
It is not clear what compound the authors are referring to. I could guess that it can be the essential oil. Since this is a mixture of at least 3 chemical compounds (limonene, citronellol, and geraniol, as the authors declare in R316) how can purity be declared based on the presence of the OH vibration bands?
This is a reformulation of the FTIR
Fourier-transform infrared (FTIR) spectroscopy was performed to analyze the chemical interactions in the pectin films, with and without EO, over a wavenumber range of 4000–600 cm⁻¹ (Figure 4). In the control film, prominent peaks were observed between 3000–3500 cm⁻¹, corresponding to O-H stretching vibrations, indicative of moisture content. The peak at 2925 cm⁻¹ represents CH₂ stretching vibrations in polysaccharides, while the peak at 1734 cm⁻¹ is associated with C=O vibrations from carboxylic acids such as glucuronic acid. Other notable peaks included those at 1603 cm⁻¹ (C=C aromatic vibrations), 1437 cm⁻¹ (C-H bending), and 1328 cm⁻¹ (C-O stretching). The FTIR spectrum of the EO-incorporated film exhibited a shift in the carbonyl peak to 1715 cm⁻¹, characteristic of C=C vibrations from compounds like limonene, citronellol, and geraniol present in lemon EO. This shift confirms the successful incorporation of EO into the pectin matrix. The spectral differences observed between the control and EO-containing films highlight the chemical interactions between pectin and EO, which enhance the film's functional properties, including its antioxidant and antimicrobial activities. Line 384-393.
Figure 3. FTIR spectroscopy of pectin film.
- Figure 3 on page 9 should have a more elaborate caption, to explain the a and b graphs content.
We have revised the figure caption to clarify the content of the graphs. The updated caption now specifies that panel A represents the control film, while panel B illustrates the film incorporated with essential oil (EO). Additionally, we have indicated that both films were analyzed using scanning electron microscopy (SEM) at magnifications of 1,000x and 500x, resulting in graphs labeled as A1 and A2 for the control film, and B1 and B2 for the EO-incorporated film.
- R328: figure 3 is not the current figure number. Rechecking is necessary.
We appreciate your observation regarding the figure number. We will recheck and ensure that Figure 3 is correctly numbered and corresponds to the appropriate content in the manuscript.
- There are many editing issues to be addressed: linked words, no space after commas or full stop points.
We will conduct a thorough review of the manuscript to address linked words, ensure proper spacing after commas and periods, and correct any other typographical errors.
- References should be cited according to the journal’s recommendations.
We acknowledge the importance of adhering to the journal’s citation guidelines. We will revise all references to ensure they are formatted according to the journal’s recommendations.
- Chemical formulas should be written with the corresponding subscripts.
We will write them.
The attached file is a highlighted version of the manuscript, for speedy revision.
Reviewer 2 Report
Comments and Suggestions for Authors
While I recognize the authors' efforts in conducting this research, I believe it doesn`t meet the standards for publication in Foods` journal. Unfortunately, the study lacks several critical analyses for the comprehensive characterization of biofilms containing essential oils, such as mechanical and physicochemical analyses (TGA, DSG, WVTR, WVP, Colour, Tensile stress and elongation, etc.), bioactive compounds profile, microbiological analyses, and additional antioxidant capacity assessments (TPC, CUPRAC, ORAC, FRAP, TEAC...). In addition, the authors only performed one type of alternative treatment, which makes it very difficult to obtain robust results that can also be compared with other studies (previous and future).
For sure, while it is compressible that not always is feasible to perform all these analyses, but the majority of them are important and provide complementary insights for this kind of paper.
Despite this, here are my recommendations to leave some contributions to the work.
- Further comparisons with other types of biodegradable films could enhance understanding of the specific advantages of this lemon EO-pectin combination.
- The study lacks an exploration of how well these films perform with food products under different environmental conditions.
- The hypothesis of using lemon EO could be more explicitly stated and connected to film properties evaluated in the present study.
- Abstract: this section needs to be revised to improve readability. Please, include more data about the specific results obtained.
- Keywords: using different keywords than the words in the paper title will help your work to be found by a wider range of researchers.
- Introduction: It would benefit from a clearer statement on the novelty and significance of using lemon EO compared to other EOs.
- MM: study lacks several critical analyses.
- Results and Discussion: linking each result back to the research objectives would improve readability. Critical comparison with findings from other similar studies, particularly regarding moisture content and antioxidant properties, are missing
The biodegradability discussion could be expanded to film durability and environmental impact. Authors should emphasize the practical significance of potential shelf-life extensions for food products when using this packaging
Ensure consistency between text and tables/figures, such as cross-referencing each table/figure in the results and confirm all figures have clear labels and legends.
Some descriptions, such as those in the FTIR and SEM sections, could be clearer by specifying the exact observations and their implications.
- Conclusion: please directly link findings to potential industry applications, such as specific food types or conditions under which these films might excel. Authors could use the conclusion to speculate on future research directions, such as testing different EOs, and discuss specific food industry applications.
- References: please, ensure to follow MDPI formatting and some recent studies (2021–2024) on biodegradable films with EOs in food packaging are missing as well lemon EO specifically.
I recommend authors to reduce the amount of wording duplication in the manuscript.
Comments on the Quality of English LanguageThe text needs to be revised and formatted correctly
Author Response
While I recognize the authors' efforts in conducting this research, I believe it doesn`t meet the standards for publication in Foods` journal. Unfortunately, the study lacks several critical analyses for the comprehensive characterization of biofilms containing essential oils, such as mechanical and physicochemical analyses (TGA, DSG, WVTR, WVP, Colour, Tensile stress and elongation, etc.), bioactive compounds profile, microbiological analyses, and additional antioxidant capacity assessments (TPC, CUPRAC, ORAC, FRAP, TEAC...). In addition, the authors only performed one type of alternative treatment, which makes it very difficult to obtain robust results that can also be compared with other studies (previous and future).
For sure, while it is compressible that not always is feasible to perform all these analyses, but the majority of them are important and provide complementary insights for this kind of paper.
Despite this, here are my recommendations to leave some contributions to the work.
- Further comparisons with other types of biodegradable films could enhance understanding of the specific advantages of this lemon EO-pectin combination.
- The study lacks an exploration of how well these films perform with food products under different environmental conditions.
- The hypothesis of using lemon EO
- Abstract: this section needs to be revised to improve readability. Please, include more data about the specific results obtained.
This study evaluated the physicochemical, morphological, and functional properties of pectin-based films incorporated with lemon essential oil (EO) to assess their potential as biodegradable food packaging materials. The results showed that EO incorporation significantly influenced the film's characteristics. The control film exhibited a smooth surface, while the EO-containing film had a rougher texture with globular structures and interconnected channels, likely representing dispersed EO droplets and matrix alterations. The mechanical analysis revealed increased elongation at break (20.05 ± 0.784%) for EO-incorporated films, indicating improved flexibility, while tensile strength and Young’s modulus decreased, suggesting reduced stiffness. Film thickness increased slightly with EO (0.097 ± 0.008 mm) compared to the control (0.089 ± 0.001 mm), though the difference was not statistically significant (p > 0.05). Moisture content decreased in EO-containing films (28.894%) compared to the control (35.236%), enhancing water resistance. Water solubility increased slightly (16.046 ± 0.003% vs. 15.315 ± 0.040%), while the swelling rate decreased significantly (0.189 ± 0.003 vs. 0.228 ± 0.040; p < 0.05), indicating greater structural stability in aqueous environments due to the hydrophobic nature of EO. Transparency tests showed that EO slightly increased film opacity (0.350 ± 0.02 vs. 0.290 ± 0.012), aligning with trends in UV-protective materials. The EO-incorporated films also exhibited moderate antibacterial activity against Staphylococcus aureus and Escherichia coli. Antifungal tests revealed strong inhibition of Botrytis cinerea (100%) and moderate inhibition of Alternaria alternata (50%) in EO-containing films. These results demonstrate that EO incorporation improves the functional properties of pectin films, enhancing their flexibility, antimicrobial activity, and environmental stability, making them promising candidates for sustainable food packaging applications. Line 17-37
- Keywords: using different keywords than the words in the paper title will help your work to be found by a wider range of researchers.
Biodegradable packaging, Pectin films, Citrus essential oils, Antioxidant activity, Active food preservation.
- Introduction: It would benefit from a clearer statement on the novelty and significance of using lemon EO compared to other EOs.
In response to the suggestion for a clearer statement on the novelty and significance of using lemon essential oil (EO) compared to other essential oils, we would like to emphasize that our choice of lemon EO is rooted in the goal of valorizing lemon peels. By extracting lemon essential oil from these by-products, we not only enhance the functional properties of pectin films but also contribute to waste reduction and sustainability in food packaging. This approach highlights the dual benefits of utilizing a natural compound with proven antioxidant properties while promoting the circular economy by making use of agricultural waste. We believe that this focus on lemon EO sets our research apart and underscores its significance in developing effective and environmentally friendly active packaging solutions. Thank you for your valuable feedback, which will help us clarify this aspect in our manuscript.
- MM: study lacks several critical analyses.
. Mechanical properties
Mechanical properties, including tensile strength, elongation at break, and Young’s modulus, were determined using a Shimadzu AGS-X universal testing machine following ASTM D882 standards (48). A 1 kN load cell was used, and tests were performed at a crosshead speed of 10 mm/min. Results were calculated as averages from five replicate measurements, with standard deviations reported.
- Antibacterial activity
The antibacterial properties of the films were tested against four bacterial strains: Staphylococcus aureus (ATCC 25923), Bacillus cereus (ATCC 6633), Escherichia coli (ATCC 25922), and Pseudomonas aeruginosa (ATCC 2785). Film discs (6 mm diameter) were cut under sterile conditions and placed on Mueller-Hinton agar plates inoculated with 10810^8108 CFU/mL of bacteria. Plates were incubated at 37°C for 18 hours. Zones of inhibition were measured to assess antibacterial activity, with ciprofloxacin serving as the positive control. Tests were performed in quintuplicate for reliability (27).
To assess the antibacterial properties of the pectin film samples, discs with a diameter of 6 mm were cut from the films by using paper puncher under sterile condition. and placed on the surface of Mueller Hinton agar plates inoculated with 0.1 mL of a bacterial culture containing approximately 108108 colony-forming units (CFU)/mL. The plates were incubated at 37°C for 18 hours, and antibacterial activity was determined by measuring the zone of inhibition around each film disc. Films without carvacrol served as negative controls. Each inhibition zone test was performed in quintuplicate to ensure the reliability of the results (27).
Ciprofloxacin was used as control positive.
Table 1. Scale for assessing antibacterial activity of the films (28).
|
Characteristics of Strain Growth |
Effect |
|
Growth inhibition zone > 1 mm around the film and under the film |
+++ Very good activity |
|
Growth inhibition zone < 1 mm around the film or only under the film |
++ Good activity |
|
No inhibition zone around the film and slight inhibition under the film |
+ Low activity |
|
No growth inhibition around or under the film |
- Lack of activity |
- Antifungal activity
The antifungal activity of pectin-based films was assessed against two phytopathogenic fungi: Alternaria alternata and Botrytis cinerea. The inhibitory effects of the films, with and without lemon essential oil (EO), on fungal mycelial growth were evaluated by measuring radial growth on Potato Dextrose Agar (PDA) medium.
A 5 mm diameter disk from a young fungal culture was aseptically placed at the center of a Petri dish containing PDA. Film samples were placed on the surface of the fungal inoculum. Dimethyl sulfoxide (DMSO) served as the control. Each treatment was conducted in quintuplicate to ensure statistical reliability (29).
The Petri dishes were incubated at 25°C for six days, after which the radial growth of fungal colonies was measured in millimeters. The percentage inhibition of mycelial growth for each fungus was calculated relative to the mean colony diameters observed in control plates. The following formula was used:
(7)
Where:
I: Inhibition rate (%)
C: Radial growth of the fungal colony in the control group (DMSO only)
T: Radial growth of the fungal colony in the presence of the tested film sample
- Results and Discussion: linking each result back to the research objectives would improve readability. Critical comparison with findings from other similar studies, particularly regarding moisture content and antioxidant properties, are missing
- Mechanical properties
The mechanical properties of the films revealed notable differences between the control film and the film with essential oil (EO) (Table 4). The control film exhibited higher tensile strength (12.28 ± 1.271 MPa), indicating a stronger, more rigid structure. In contrast, the film with EO demonstrated a slightly higher elongation at break (20.05 ± 0.784%) and a lower Young’s modulus (174.73 ± 1.915 MPa), suggesting enhanced flexibility and reduced stiffness. These changes are likely due to the incorporation of EO, which may act as a plasticizer, reducing polymer chain interactions within the pectin matrix.
The tensile strength of the film with EO was comparable to findings by Jiang et al. (32), who reported tensile strength values of 31.26 ± 2.30 MPa for lemon peel pectin films. Additionally, the observed elongation aligns with studies on mandarin peel pectin films, where elongation values reached 17.26 ± 1.67% (41). The results are consistent with Baghi et al. (42), who reported a decrease in tensile strength and Young’s modulus after incorporating nanoemulsions into pectin films, with a corresponding increase in elongation at break. However, the observed trends deviate from Vahedikia et al. (43), who found that incorporating cinnamon essential oil with chitosan nanoparticles (CEO-CNPs) increased tensile strength but reduced elongation in zein film composites. These differences may be attributed to the nature of the essential oils, film composition, and interaction between EO and the polymer matrix.
The mechanical properties of the EO-incorporated films suggest their suitability for flexible packaging applications. However, their reduced stiffness may limit their use in applications requiring high mechanical strength.
Table 4. Mechanical properties of pectin films (Mean ± SD) .
|
Film Sample |
Elongation (%) |
Strength (MPa) |
Young's Modulus (MPa) |
|
Control Film |
11.52 ± 0.123 |
12.28 ± 1.271 |
494.77 ± 0.519 |
|
Film with EO |
20.05 ± 0.784 |
14.68 ± 1.419 |
174.73 ± 1.915 |
- Antibacterial activity
The antibacterial efficacy of the films was tested against Gram-positive bacteria (Staphylococcus aureus and Bacillus cereus) and Gram-negative bacteria (Escherichia coli and Pseudomonas aeruginosa) (Table 5, Figure 6). The control film exhibited no antibacterial activity, with zero zones of inhibition for all bacterial strains. Conversely, the film with EO showed moderate antibacterial activity, with a zone of inhibition of 0.930 ± 0.574 mm against Staphylococcus aureus and 0.750 ± 0.203 mm against Escherichia coli. However, no inhibition was observed for Bacillus cereus or Pseudomonas aeruginosa, suggesting that the efficacy of the EO varies depending on the bacterial strain. These results align with studies highlighting the variability in antibacterial effectiveness of essential oils due to their chemical composition. Gram-negative bacteria, such as E. coli and Pseudomonas aeruginosa, are more resistant to EO due to the presence of hydrophobic lipopolysaccharides in their outer membrane, which limit the diffusion of hydrophobic compounds (44; 45). In contrast, Gram-positive bacteria like S. aureus lack this outer membrane, allowing essential oils to penetrate more effectively through their peptidoglycan cell wall. These findings are consistent with Bharti et al. (46), who reported that starch bio-based composite edible films functionalized with Carum carvi L. essential oil showed greater antibacterial activity against Gram-positive bacteria compared to Gram-negative bacteria. The enhanced antibacterial properties of the EO-incorporated films indicate their potential for use in food packaging to enhance safety and extend shelf life.
Table 5. Antibacterial activity of the films (Mean ± SD) .
|
Film |
Staphylococcus aureus (mm) |
Escherichia coli (mm) |
Bacillus cereus (mm) |
Pseudomonas aeruginosa (mm) |
||||
|
Control film |
− |
− |
− |
− |
− |
− |
− |
− |
|
Film with EO |
0.930± 0.574 |
++ |
− |
− |
0.750± 0.203 |
++ |
− |
− |
++ = Moderate antibacterial activity
- = No inhibition
Figure 6. Antibacterial activity of the pectin films, A is the control film, B is the film with EO.
- Antifungal activity
The antifungal activity of the pectin films was assessed against Alternaria alternata and Botrytis cinerea, as shown in Figure 7 and summarized in Table 6. The antifungal properties of these films are critical for their potential use in food packaging, where they can enhance the safety and preservation of food products. The results showed that the films with essential oils (EO) exhibited stronger inhibition against Botrytis cinerea, with 100% inhibition for the film containing lemon EO compared to only 70% inhibition by the control film. On the other hand, when tested against Alternaria alternata, the control film showed a modest 10% inhibition, while the film with EO demonstrated a 50% inhibition rate. These results suggest that pectin films, particularly those incorporated with EO, possess inherent antifungal properties, especially against Botrytis cinerea.
These findings align with previous studies, such as Alvarez et al. (47) investigated natural pectin-based edible coatings with antifungal properties aimed at controlling green mold in ‘Valencia’ oranges. Their study showed that these coatings successfully reduced postharvest losses and maintained fruit quality by inhibiting fungal growth, highlighting the potential of pectin films as an effective and sustainable approach for managing postharvest diseases in citrus fruits.
Table 6. Antifungal activity of pectin films(Mean ± SD) .
|
Film samples |
Inhibition (%) Botrytis cinerea |
Inhibition (%) Alternaria alternata |
|
Control film |
70± 2.500 |
10±4.800 |
|
Film with EO |
100 ± 0.000 |
50 ± 1.700 |
Figure 7. Antifungal activity of pectin films a1- Alternaria alternata with DMSO, a2 Alternaria alternata control film, a3 Alternaria alternata film with EO, b1 Botrytis cinerea with DMSO, b2 Botrytis cinerea control film, b3 Botrytis cinerea film with EO.
The biodegradability discussion could be expanded to film durability and environmental impact. Authors should emphasize the practical significance of potential shelf-life extensions for food products when using this packaging
This reduction in biodegradation not only highlights the potential for extending the shelf life of food products packaged in these films but also emphasizes their practical significance in reducing food waste. By enhancing the durability of pectin films through the incorporation of EO, we can provide effective packaging solutions that maintain food quality for longer periods, thereby minimizing environmental impact. Furthermore, this approach aligns with sustainable practices by valorizing lemon peels and utilizing natural compounds, contributing to a circular economy.
Ensure consistency between text and tables/figures, such as cross-referencing each table/figure in the results and confirm all figures have clear labels and legends.
We appreciate your attention to this important detail. We will conduct a thorough review of the manuscript to ensure that all tables and figures are properly cross-referenced in the results section. Additionally, we will verify that each figure has clear labels and legends that accurately describe the content and findings presented. This will enhance the clarity and coherence of our presentation, making it easier for readers to understand the relationships between the text and visual data.
Some descriptions, such as those in the FTIR and SEM sections, could be clearer by specifying the exact observations and their implications.
- Conclusion: please directly link findings to potential industry applications, such as specific food types or conditions under which these films might excel. Authors could use the conclusion to speculate on future research directions, such as testing different EOs, and discuss specific food industry applications.
We would like to highlight that we are currently in the process of writing an article focused on the application of pectin films in the food industry. This upcoming work will discuss specific food types and conditions under which these films may excel, emphasizing their practical significance in extending shelf life and enhancing food safety. By utilizing lemon essential oil in pectin films, we aim to create active packaging solutions that can be particularly effective for perishable products such as fruits, vegetables, and dairy items. The antioxidant properties of the essential oil can help preserve the quality of these foods while minimizing spoilage. Additionally, we will explore future research directions, including testing different essential oils to further enhance the functionality of pectin films and their applicability across various food products.
- References: please, ensure to follow MDPI formatting and some recent studies (2021–2024) on biodegradable films with EOs in food packaging are missing as well lemon EO specifically.
We will ensure that our manuscript is formatted according to MDPI's specific guidelines, as outlined in their style guide.
I recommend authors to reduce the amount of wording duplication in the manuscript.
While I recognize the authors' efforts in conducting this research, I believe it doesn`t meet the standards for publication in Foods` journal. Unfortunately, the study lacks several critical analyses for the comprehensive characterization of biofilms containing essential oils, such as mechanical and physicochemical analyses (TGA, DSG, WVTR, WVP, Colour, Tensile stress and elongation, etc.), bioactive compounds profile, microbiological analyses, and additional antioxidant capacity assessments (TPC, CUPRAC, ORAC, FRAP, TEAC...). In addition, the authors only performed one type of alternative treatment, which makes it very difficult to obtain robust results that can also be compared with other studies (previous and future).
For sure, while it is compressible that not always is feasible to perform all these analyses, but the majority of them are important and provide complementary insights for this kind of paper.
Despite this, here are my recommendations to leave some contributions to the work.
- Further comparisons with other types of biodegradable films could enhance understanding of the specific advantages of this lemon EO-pectin combination.
- The study lacks an exploration of how well these films perform with food products under different environmental conditions.
- The hypothesis of using lemon EO
- Abstract: this section needs to be revised to improve readability. Please, include more data about the specific results obtained.
This study evaluated the physicochemical, morphological, and functional properties of pectin-based films incorporated with lemon essential oil (EO) to assess their potential as biodegradable food packaging materials. The results showed that EO incorporation significantly influenced the film's characteristics. The control film exhibited a smooth surface, while the EO-containing film had a rougher texture with globular structures and interconnected channels, likely representing dispersed EO droplets and matrix alterations. The mechanical analysis revealed increased elongation at break (20.05 ± 0.784%) for EO-incorporated films, indicating improved flexibility, while tensile strength and Young’s modulus decreased, suggesting reduced stiffness. Film thickness increased slightly with EO (0.097 ± 0.008 mm) compared to the control (0.089 ± 0.001 mm), though the difference was not statistically significant (p > 0.05). Moisture content decreased in EO-containing films (28.894%) compared to the control (35.236%), enhancing water resistance. Water solubility increased slightly (16.046 ± 0.003% vs. 15.315 ± 0.040%), while the swelling rate decreased significantly (0.189 ± 0.003 vs. 0.228 ± 0.040; p < 0.05), indicating greater structural stability in aqueous environments due to the hydrophobic nature of EO. Transparency tests showed that EO slightly increased film opacity (0.350 ± 0.02 vs. 0.290 ± 0.012), aligning with trends in UV-protective materials. The EO-incorporated films also exhibited moderate antibacterial activity against Staphylococcus aureus and Escherichia coli. Antifungal tests revealed strong inhibition of Botrytis cinerea (100%) and moderate inhibition of Alternaria alternata (50%) in EO-containing films. These results demonstrate that EO incorporation improves the functional properties of pectin films, enhancing their flexibility, antimicrobial activity, and environmental stability, making them promising candidates for sustainable food packaging applications. Line 17-37
- Keywords: using different keywords than the words in the paper title will help your work to be found by a wider range of researchers.
Biodegradable packaging, Pectin films, Citrus essential oils, Antioxidant activity, Active food preservation.
- Introduction: It would benefit from a clearer statement on the novelty and significance of using lemon EO compared to other EOs.
In response to the suggestion for a clearer statement on the novelty and significance of using lemon essential oil (EO) compared to other essential oils, we would like to emphasize that our choice of lemon EO is rooted in the goal of valorizing lemon peels. By extracting lemon essential oil from these by-products, we not only enhance the functional properties of pectin films but also contribute to waste reduction and sustainability in food packaging. This approach highlights the dual benefits of utilizing a natural compound with proven antioxidant properties while promoting the circular economy by making use of agricultural waste. We believe that this focus on lemon EO sets our research apart and underscores its significance in developing effective and environmentally friendly active packaging solutions. Thank you for your valuable feedback, which will help us clarify this aspect in our manuscript.
- MM: study lacks several critical analyses.
. Mechanical properties
Mechanical properties, including tensile strength, elongation at break, and Young’s modulus, were determined using a Shimadzu AGS-X universal testing machine following ASTM D882 standards (48). A 1 kN load cell was used, and tests were performed at a crosshead speed of 10 mm/min. Results were calculated as averages from five replicate measurements, with standard deviations reported.
- Antibacterial activity
The antibacterial properties of the films were tested against four bacterial strains: Staphylococcus aureus (ATCC 25923), Bacillus cereus (ATCC 6633), Escherichia coli (ATCC 25922), and Pseudomonas aeruginosa (ATCC 2785). Film discs (6 mm diameter) were cut under sterile conditions and placed on Mueller-Hinton agar plates inoculated with 10810^8108 CFU/mL of bacteria. Plates were incubated at 37°C for 18 hours. Zones of inhibition were measured to assess antibacterial activity, with ciprofloxacin serving as the positive control. Tests were performed in quintuplicate for reliability (27).
To assess the antibacterial properties of the pectin film samples, discs with a diameter of 6 mm were cut from the films by using paper puncher under sterile condition. and placed on the surface of Mueller Hinton agar plates inoculated with 0.1 mL of a bacterial culture containing approximately 108108 colony-forming units (CFU)/mL. The plates were incubated at 37°C for 18 hours, and antibacterial activity was determined by measuring the zone of inhibition around each film disc. Films without carvacrol served as negative controls. Each inhibition zone test was performed in quintuplicate to ensure the reliability of the results (27).
Ciprofloxacin was used as control positive.
Table 1. Scale for assessing antibacterial activity of the films (28).
|
Characteristics of Strain Growth |
Effect |
|
Growth inhibition zone > 1 mm around the film and under the film |
+++ Very good activity |
|
Growth inhibition zone < 1 mm around the film or only under the film |
++ Good activity |
|
No inhibition zone around the film and slight inhibition under the film |
+ Low activity |
|
No growth inhibition around or under the film |
- Lack of activity |
- Antifungal activity
The antifungal activity of pectin-based films was assessed against two phytopathogenic fungi: Alternaria alternata and Botrytis cinerea. The inhibitory effects of the films, with and without lemon essential oil (EO), on fungal mycelial growth were evaluated by measuring radial growth on Potato Dextrose Agar (PDA) medium.
A 5 mm diameter disk from a young fungal culture was aseptically placed at the center of a Petri dish containing PDA. Film samples were placed on the surface of the fungal inoculum. Dimethyl sulfoxide (DMSO) served as the control. Each treatment was conducted in quintuplicate to ensure statistical reliability (29).
The Petri dishes were incubated at 25°C for six days, after which the radial growth of fungal colonies was measured in millimeters. The percentage inhibition of mycelial growth for each fungus was calculated relative to the mean colony diameters observed in control plates. The following formula was used:
(7)
Where:
I: Inhibition rate (%)
C: Radial growth of the fungal colony in the control group (DMSO only)
T: Radial growth of the fungal colony in the presence of the tested film sample
- Results and Discussion: linking each result back to the research objectives would improve readability. Critical comparison with findings from other similar studies, particularly regarding moisture content and antioxidant properties, are missing
- Mechanical properties
The mechanical properties of the films revealed notable differences between the control film and the film with essential oil (EO) (Table 4). The control film exhibited higher tensile strength (12.28 ± 1.271 MPa), indicating a stronger, more rigid structure. In contrast, the film with EO demonstrated a slightly higher elongation at break (20.05 ± 0.784%) and a lower Young’s modulus (174.73 ± 1.915 MPa), suggesting enhanced flexibility and reduced stiffness. These changes are likely due to the incorporation of EO, which may act as a plasticizer, reducing polymer chain interactions within the pectin matrix.
The tensile strength of the film with EO was comparable to findings by Jiang et al. (32), who reported tensile strength values of 31.26 ± 2.30 MPa for lemon peel pectin films. Additionally, the observed elongation aligns with studies on mandarin peel pectin films, where elongation values reached 17.26 ± 1.67% (41). The results are consistent with Baghi et al. (42), who reported a decrease in tensile strength and Young’s modulus after incorporating nanoemulsions into pectin films, with a corresponding increase in elongation at break. However, the observed trends deviate from Vahedikia et al. (43), who found that incorporating cinnamon essential oil with chitosan nanoparticles (CEO-CNPs) increased tensile strength but reduced elongation in zein film composites. These differences may be attributed to the nature of the essential oils, film composition, and interaction between EO and the polymer matrix.
The mechanical properties of the EO-incorporated films suggest their suitability for flexible packaging applications. However, their reduced stiffness may limit their use in applications requiring high mechanical strength.
Table 4. Mechanical properties of pectin films (Mean ± SD) .
|
Film Sample |
Elongation (%) |
Strength (MPa) |
Young's Modulus (MPa) |
|
Control Film |
11.52 ± 0.123 |
12.28 ± 1.271 |
494.77 ± 0.519 |
|
Film with EO |
20.05 ± 0.784 |
14.68 ± 1.419 |
174.73 ± 1.915 |
- Antibacterial activity
The antibacterial efficacy of the films was tested against Gram-positive bacteria (Staphylococcus aureus and Bacillus cereus) and Gram-negative bacteria (Escherichia coli and Pseudomonas aeruginosa) (Table 5, Figure 6). The control film exhibited no antibacterial activity, with zero zones of inhibition for all bacterial strains. Conversely, the film with EO showed moderate antibacterial activity, with a zone of inhibition of 0.930 ± 0.574 mm against Staphylococcus aureus and 0.750 ± 0.203 mm against Escherichia coli. However, no inhibition was observed for Bacillus cereus or Pseudomonas aeruginosa, suggesting that the efficacy of the EO varies depending on the bacterial strain. These results align with studies highlighting the variability in antibacterial effectiveness of essential oils due to their chemical composition. Gram-negative bacteria, such as E. coli and Pseudomonas aeruginosa, are more resistant to EO due to the presence of hydrophobic lipopolysaccharides in their outer membrane, which limit the diffusion of hydrophobic compounds (44; 45). In contrast, Gram-positive bacteria like S. aureus lack this outer membrane, allowing essential oils to penetrate more effectively through their peptidoglycan cell wall. These findings are consistent with Bharti et al. (46), who reported that starch bio-based composite edible films functionalized with Carum carvi L. essential oil showed greater antibacterial activity against Gram-positive bacteria compared to Gram-negative bacteria. The enhanced antibacterial properties of the EO-incorporated films indicate their potential for use in food packaging to enhance safety and extend shelf life.
Table 5. Antibacterial activity of the films (Mean ± SD) .
|
Film |
Staphylococcus aureus (mm) |
Escherichia coli (mm) |
Bacillus cereus (mm) |
Pseudomonas aeruginosa (mm) |
||||
|
Control film |
− |
− |
− |
− |
− |
− |
− |
− |
|
Film with EO |
0.930± 0.574 |
++ |
− |
− |
0.750± 0.203 |
++ |
− |
− |
++ = Moderate antibacterial activity
- = No inhibition
Figure 6. Antibacterial activity of the pectin films, A is the control film, B is the film with EO.
- Antifungal activity
The antifungal activity of the pectin films was assessed against Alternaria alternata and Botrytis cinerea, as shown in Figure 7 and summarized in Table 6. The antifungal properties of these films are critical for their potential use in food packaging, where they can enhance the safety and preservation of food products. The results showed that the films with essential oils (EO) exhibited stronger inhibition against Botrytis cinerea, with 100% inhibition for the film containing lemon EO compared to only 70% inhibition by the control film. On the other hand, when tested against Alternaria alternata, the control film showed a modest 10% inhibition, while the film with EO demonstrated a 50% inhibition rate. These results suggest that pectin films, particularly those incorporated with EO, possess inherent antifungal properties, especially against Botrytis cinerea.
These findings align with previous studies, such as Alvarez et al. (47) investigated natural pectin-based edible coatings with antifungal properties aimed at controlling green mold in ‘Valencia’ oranges. Their study showed that these coatings successfully reduced postharvest losses and maintained fruit quality by inhibiting fungal growth, highlighting the potential of pectin films as an effective and sustainable approach for managing postharvest diseases in citrus fruits.
Table 6. Antifungal activity of pectin films(Mean ± SD) .
|
Film samples |
Inhibition (%) Botrytis cinerea |
Inhibition (%) Alternaria alternata |
|
Control film |
70± 2.500 |
10±4.800 |
|
Film with EO |
100 ± 0.000 |
50 ± 1.700 |
Figure 7. Antifungal activity of pectin films a1- Alternaria alternata with DMSO, a2 Alternaria alternata control film, a3 Alternaria alternata film with EO, b1 Botrytis cinerea with DMSO, b2 Botrytis cinerea control film, b3 Botrytis cinerea film with EO.
The biodegradability discussion could be expanded to film durability and environmental impact. Authors should emphasize the practical significance of potential shelf-life extensions for food products when using this packaging
This reduction in biodegradation not only highlights the potential for extending the shelf life of food products packaged in these films but also emphasizes their practical significance in reducing food waste. By enhancing the durability of pectin films through the incorporation of EO, we can provide effective packaging solutions that maintain food quality for longer periods, thereby minimizing environmental impact. Furthermore, this approach aligns with sustainable practices by valorizing lemon peels and utilizing natural compounds, contributing to a circular economy.
Ensure consistency between text and tables/figures, such as cross-referencing each table/figure in the results and confirm all figures have clear labels and legends.
We appreciate your attention to this important detail. We will conduct a thorough review of the manuscript to ensure that all tables and figures are properly cross-referenced in the results section. Additionally, we will verify that each figure has clear labels and legends that accurately describe the content and findings presented. This will enhance the clarity and coherence of our presentation, making it easier for readers to understand the relationships between the text and visual data.
Some descriptions, such as those in the FTIR and SEM sections, could be clearer by specifying the exact observations and their implications.
- Conclusion: please directly link findings to potential industry applications, such as specific food types or conditions under which these films might excel. Authors could use the conclusion to speculate on future research directions, such as testing different EOs, and discuss specific food industry applications.
We would like to highlight that we are currently in the process of writing an article focused on the application of pectin films in the food industry. This upcoming work will discuss specific food types and conditions under which these films may excel, emphasizing their practical significance in extending shelf life and enhancing food safety. By utilizing lemon essential oil in pectin films, we aim to create active packaging solutions that can be particularly effective for perishable products such as fruits, vegetables, and dairy items. The antioxidant properties of the essential oil can help preserve the quality of these foods while minimizing spoilage. Additionally, we will explore future research directions, including testing different essential oils to further enhance the functionality of pectin films and their applicability across various food products.
- References: please, ensure to follow MDPI formatting and some recent studies (2021–2024) on biodegradable films with EOs in food packaging are missing as well lemon EO specifically.
We will ensure that our manuscript is formatted according to MDPI's specific guidelines, as outlined in their style guide.
I recommend authors to reduce the amount of wording duplication in the manuscript.
Reviewer 3 Report
Comments and Suggestions for Authors
I do not think the paper can be accepted at current stage. Some comments are list below:
1 The refferences are too old, it should be improved.
2 The novelty of the paper compared with the previous work should be pointed out in the section of Introduction and the section should be improved.
3 I think, the effectiveness of these films as an efficient active packaging for food applications should be investigated in this paper, not in the future! So, experiment on this topic should be added in the reversed manuscript.
Author Response
I do not think the paper can be accepted at current stage. Some comments are list below:
1 The references are too old; it should be improved.
We appreciate your feedback and have incorporated more recent studies to enhance the relevance and timeliness of our literature review. Below are updated references that reflect recent advancements in the field
2 The novelty of the paper compared with the previous work should be pointed out in the section of Introduction and the section should be improved.
The development of active packaging, which integrates bioactive compounds into packaging materials, has gained significant traction in recent years due to its ability to preserve food by preventing microbial contamination and oxidative damage. This innovative approach enhances food safety and shelf life while meeting consumer demand for sustainable, minimally processed, and additive-free products (1;2). The main role of active packaging is the preservation of food from any microbial contamination and oxidative stress. The use of this packaging by consumers and industries has experienced a real boom in recent years due to its biological origin, above all its properties, especially to improve food safety and shelf life without causing harmful effects. Because of worries about the harmful impact of synthetic additives on human health, modern consumers prefer foods with little to no artificial additions (3; 4).
An alternate supply for natural components used in the production of bio-based packaging is waste and byproducts (5). Given their abundance of functional ingredients (vitamin C, fibers, carotenoids, essential oils, and phenolic compounds) and their many uses in the food, cosmetics, nutraceutical, biofuel, and materials production industries, citrus peels are a highly valuable matrix (6). Citrus fruits are renowned for their medicinal and therapeutic properties, largely attributed to their rich array of bioactive components. Comprising 75–90% water, 6–9% sugars, and the remainder consisting of pectin, dietary fiber, minerals, and essential oils, citrus fruits offer a healthy and balanced dietary option. Additionally, they are a significant source of carotenoids and flavonoids (7). According to FAO STAT (8), global citrus fruit production has significantly increased over the years, rising from approximately 51.48 million tons (MT) in 1975 to 158 MT in 2019. However, the waste generated from citrus fruit processing presents an opportunity to harness valuable bioactive constituents such as phenols, flavonoids, carotenoids, and essential oils. Furthermore, in the context of bioeconomy, this waste could be repurposed for the production of biofuels and other valuable commodities (9).
Self. Pectin, a water-soluble anionic heteropolysaccharide of plant origin, is widely used in the food industry, mainly as a gelling, thickening, and stabilizing agent in fruit products (10). It’s extracted from citrus fruits, apples, currants, black currants, quinces…etc. While pectin can produce packaging films with excellent mechanical properties and barrier functions, there is still a need to enhance the functionalities of pectin-based films by incorporating additional bioactive compounds. This enhancement aims to improve their ability to protect food products and extend their shelf life. The inclusion of active compounds in packaging not only boosts the functional properties of the films but also contributes to prolonging the freshness and quality of the packaged products (11).
Essential oils (EO) are a good example of natural compounds obtained from a wide variety of plant aromatics that have been applied in the manufacture of food packaging (12). However, these substances have been applied to low-cost non-biodegradable polymers and biopolymers whose treatment is complex (13).Waste recovery not only reduces the ecological impact by minimizing pollution but also offers new opportunities allowing economic development; furthermore, the recycling of waste from the agro-processing industry not only minimizes pollution and reduces ecological impact but also offers new opportunities to enable sustainable economic development in many sectors (14).
However, not all of them are appropriate for use in food packaging applications because of their migration qualities, volatility, and unfavorable taste and/or odor (15). According to Rehman et al. (16), the primary issue with directly adding EO to active packaging is the migration of active compounds, which further decreases the efficiency of active packaging during the food's shelf life. Novel encapsulation techniques play a crucial role in enhancing the stability of bioactive compounds, such as essential oils and plant extracts, while minimizing their reactivity and losses. These encapsulation strategies effectively shield bioactive from external factors that can lead to degradation, thereby preserving their functional properties. By incorporating bioactive compounds into protective matrices, encapsulation not only improves their stability but also controls their release, enhances solubility, and increases bioavailability (17).
In this context, the present study aims to achieve a double valorization of lemon peels by extracting essential oil, pectin. It further focuses on developing bioactive pectin-based films incorporating Citrus limon essential oil and evaluating their mechanical, antioxidant, and physical properties. This work not only contributes to waste valorization but also offers a sustainable approach to the production of bioactive packaging materials with potential applications in the food industry.
Line 31-100
3 I think, the effectiveness of these films as an efficient active packaging for food applications should be investigated in this paper, not in the future! So, experiment on this topic should be added in the reversed manuscript.
We would like to highlight that we are currently in the process of writing an article focused on the application of pectin films in the food industry. This upcoming work will discuss specific food types and conditions under which these films may excel, emphasizing their practical significance in extending shelf life and enhancing food safety. By utilizing lemon essential oil in pectin films, we aim to create active packaging solutions that can be particularly effective for perishable products such as fruits, vegetables, and dairy items. The antioxidant properties of the essential oil can help preserve the quality of these foods while minimizing spoilage. Additionally, we will explore future research directions, including testing different essential oils to further enhance the functionality of pectin films and their applicability across various food products.
Reviewer 4 Report
Comments and Suggestions for Authors
This paper studies the citrus lemon essential oil mixed in the pectin film, and determines its preparation method, physical properties, antioxidant properties, etc., which provides a direction for the study of the effectiveness of food active packaging. The logic of the article is clear, but there are still some problems:
1. There are several detailed errors, please check and correct them carefully. For example, the 71st line "sing" is changed to "Saying"; line 98 "48hr" is changed to "48h"; line 102 "CaC12" is changed to "CaCl"; line 116 "105" should be added units; line 168 "theblank" has no space in the middle; line 174 "K2S204" is changed to "K2S 20"; "Ca2+" in line 228 should be changed to "Ca*"; "," in Table 3 should be changed to ""; the format of "citruslimon" in lines 330 and 332 is inconsistent.
2. Some grammars in the text are inappropriate. For example, "in to reactor" in line 8I; "25°C and 80°C baths" in line 126 is unknown; "suggests" in line 252 should use the past tense, etc.
3. In Part a of Chapter 3, you can supplement in detail how the film preparation technology and drying conditions affect the thickness of the film.
4.The enumerable examples in Part C of Chapter 3 illustrate how the addition of essential oil causes changes in the structure and thickness of the film.
5. Part d of Chapter 3 is logically confused when describing the relationship between essential oil addition and pectin transparency. Please check again.
6. At the end of the h part of Chapter 3, some other research examples can be added to prove that these particles are caused by changes in pectin molecules during film formation.
7. The format of the English reference is wrong, please check it carefully.
Author Response
This paper studies the citrus lemon essential oil mixed in the pectin film, and determines its preparation method, physical properties, antioxidant properties, etc., which provides a direction for the study of the effectiveness of food active packaging. The logic of the article is clear, but there are still some problems:
- There are several detailed errors, please check and correct them carefully. For example, the 71st line "sing" is changed to "Saying"; line 98 "48hr" is changed to "48h"; line 102 "CaC12" is changed to "CaCl"; line 116 "105" should be added units; line 168 "theblank" has no space in the middle; line 174 "K2S204" is changed to "K2S 20"; "Ca2+" in line 228 should be changed to "Ca*"; "," in Table 3 should be changed to ""; the format of "citruslimon" in lines 330 and 332 is inconsistent.
We sincerely appreciate your attention to detail. We have carefully reviewed and corrected them.
- Some grammars in the text are inappropriate. For example, "in to reactor" in line 8I; "25°C and 80°C baths" in line 126 is unknown; "suggests" in line 252 should use the past tense, etc.
We sincerely appreciate your attention to detail. We have carefully reviewed and corrected them
- In Part a of Chapter 3, you can supplement in detail how the film preparation technology and drying conditions affect the thickness of the film.
It is generally influenced by the preparation techniques and drying conditions (31). Jiang et al. (32) noted that an increase in solid content in the film-forming solution results in thicker films, as observed in this study. Line 295-297.
4.The enumerable examples in Part C of Chapter 3 illustrate how the addition of essential oil causes changes in the structure and thickness of the film.
Furlan et al. (33) also observed structural and thickness modifications in pectin films incorporated with thyme essential oil, supporting these findings. Line 297-298.
- Part d of Chapter 3 is logically confused when describing the relationship between essential oil addition and pectin transparency. Please check again.
Film transparency is another vital property, particularly for food packaging, as it can affect the appearance of packaged products and offer protection against UV radiation. The opacity of films with and without EO was measured and found to be 0.350±0.02 and 0.290±0.12, respectively (Table 2). While the difference was not statistically significant (p≥0.05), the trend aligns with previous studies indicating that EO incorporation often increases film opacity. Line 337-341.
- At the end of the h part of Chapter 3, some other research examples can be added to prove that these particles are caused by changes in pectin molecules during film formation.
The surface morphology of pectin films with and without essential oil (EO) was analyzed using SEM micrographs (Figure 5). The control film exhibited a smooth surface with minor variations, indicating a network of fine, thread-like structures that likely represent the pectin biopolymer. This smooth and uniform appearance is characteristic of pure pectin films, suggesting a cohesive polymer network. In contrast, films incorporated with EO displayed a rougher surface texture, characterized by globular structures that are likely EO droplets dispersed within the pectin matrix. Furthermore, interconnected channels were observed throughout the film, which could have resulted from the interaction and incorporation of EO into the biopolymer. Both film types exhibited a grainy structure, potentially caused by molecular changes in the pectin during film formation. These findings are consistent with previous studies. Nisar et al. (40) reported that the incorporation of chitosan nanoparticles into pectin films led to significant morphological changes, including a more complex microstructure. Similarly, Furlan et al. (33) observed that the addition of thyme essential oil to pectin films altered the surface morphology, leading to increased roughness due to interactions between the EO and the polymer chains. These morphological differences highlight the impact of EO addition on the structural characteristics of pectin films. The changes observed could influence the functional properties of the films, such as barrier performance and mechanical behavior, which are crucial for their application in food packaging. Line 401-419.
- The format of the English reference is wrong, please check it carefully.
We sincerely appreciate your attention to detail. We have carefully reviewed and corrected the references to ensure they adhere to MDPI style. Below is the revised list of references:
(1) Realini, C. E.; Marcos, B. Active and intelligent packaging systems for a modern society. Meat Science 2014, 98(3), 404–419. doi: 10.1016/j.meatsci.2014.05.022.
(2) QIAN, Mengyan, LIU, Donghong, ZHANG, Xinhui, et al. A review of active packaging in bakery products: Applications and future trends. Trends in Food Science & Technology, 2021, vol. 114, p. 459-471.
(3) ALIZADEH-SANI, Mahmood, MOHAMMADIAN, Esmail, RHIM, Jong-Whan, et al. pH-sensitive (halochromic) smart packaging films based on natural food colorants for the monitoring of food quality and safety. Trends in Food Science & Technology, 2020, vol. 105, p. 93-144.
(4) SANI, Mahmood Alizadeh, AZIZI-LALABADI, Maryam, TAVASSOLI, Milad, et al. Recent advances in the development of smart and active biodegradable packaging materials. Nanomaterials, 2021, vol. 11, no 5, p. 1331.
(5) Adilah, A.N.; Jamilah, B.; Noranizan, M.A.; Hanani, Z.N. Utilization of Mango Peel Extracts on the Biodegradable Films for Active Packaging. Food Packag. Shelf Life 2018, 16, 1-7. doi: 10.1016/j.fpsl.2018.04.001.
(6) Ledesma-Escobar, C.A.; de Castro, M.D.L. Towards a comprehensive exploitation of citrus. Trends in Food Science & Technology 2014, 39(1), 63-75.doi: https://doi.org/10.1016/j.tifs.2014.06.003
(7) SANTIAGO, B., MOREIRA, M. T., FEIJOO, G., & GONZALEZ-GARCIA, S. Identification of environmental aspects of citrus waste valorization into D-limonene from a biorefinery approach. Biomass and Bioenergy, 2020, vol. 143, p. 105844.
(8) FAO STAT. Food and Agriculture Data. Food and Agriculture Organization. FAO, Rome. Available from: http://www.fao.org/faostat/en/?#data/QC (2019).
(9) SURI, S., SINGH, A., & NEMA, P. K. Recent advances in valorization of citrus fruits processing waste: A way forward towards environmental sustainability. Food Science and Biotechnology, 2021, p. 1-26.
(10) Bouziane, H. ; Maachou, H. ; Zouambia, Y. Contribution à l’élaboration de nouveaux systèmes de libération des médicaments à base de composites pectine-chitosane. J. Food Sci. 2016.
(11) DIRPAN, Andi, DELIANA, Yosini, AINANI, Andi Fadiah, et al. Exploring the Potential of Pectin as a Source of Biopolymers for Active and Intelligent Packaging: A Review. Polymers, 2024, vol. 16, no 19, p. 2783.
(12) BHARADVAJA, Navneeta. Aromatic plants: a multifaceted asset. Brazilian Journal of Botany, 2023, vol. 46, no 2, p. 241-254.
(13) Yahyaoui, M. Application des huiles essentielles dans le domaine des emballages.
(14) M'Hiri, N. Étude comparative de l’effet des méthodes d’extraction sur les phénols et l’activité antioxydante des extraits des écorces « Maltaise demi sanguine » et exploration de l’effet inhibiteur de la corrosion de l’acier au carbone. Univ. Carthage, Tunisie 2015, pp. 1-2-3-10-11.
(15) Herrera, P.; Aydin, M.; Park, S.H.; Khatiwara, A.; Ahn, S. Utility of egg yolk antibodies for detection and control of foodborne Salmonella. Agric. Food Anal. Bacteriol. 2013, 3, 195-217.
(16) Rehman, A.; Jafari, S. M.; Aadil, R. M.; Assadpour, E.; Randhawa, M. A.; Mahmood, S. Development of active food packaging via incorporation of biopolymeric nanocarriers containing essential oils. Trends in Food Science & Technology 2020, 101, 106-121. doi: 10.1016/j.tifs.2020.05.014.
(17) Zabot, G.L.; Rodrigues, F.S.; Ody, L.P.; Vin, M.; Herrera, E.; Palacin, H.; Javier, S.C.; Best, I.; Olivera-montenegro, L. Encapsulation of Bioactive Compounds for Food and Agricultural Applications. Polymers 2022, 14, 4194.
(18) European Pharmacopoeia 10.0. Council of Europe, Strasbourg Cedex, France, 2019.
(19) Jamdar, F.; Ali Mortazavi, S.; Reza Saiedi Asl, M.; Sharifi, A. Physicochemical properties and enzymatic activity of wheat germ extract microencapsulated with spray and freeze drying. Food Science & Nutrition 2021, 9(2), 1192-1201.
(20) Wang, Q.; Tian, F.; Feng, Z.; Fan, X.; Pan, Z.; Zhou, J. Antioxidant activity and physicochemical properties of chitosan films incorporated with Lycium barbarum fruit extract for active food packaging. International Journal of Food Science & Technology 2015, 50(2), 458-464. doi:10.1111/ijfs.12659
(21) Ahmed, S.; Ikram, S. Chitosan and Gelatin Based Biodegradable Packaging Films with UV-Light Protection. J. Photochem. Photobiol. B Biol. 2016, 163, 115–124. doi: 10.1016/j.jphotobiol.2016.05.002.
(22) ZAREEN, Saba, NAWAB, Anjum, ALAM, Feroz, et al. The impact of heat‐moisture treatment and pre‐gelatinization on functional, mechanical, and barrier properties of basmati rice starch films. Cereal Chemistry, 2023, vol. 100, no 3, p. 675-684.
(23) Wu, C. S. Preparation, characterization, and biodegradability of renewable resource-based composites from recycled polylactide bioplastic and sisal fibers. Journal of Applied Polymer Science 2012, 123(1), 347-355. doi:10.1002/app.34656.
(24) CUI, Yuqian, CHENG, Yixiu, XU, Zhan, et al. Cellulose‐Based Transparent Edible Antibacterial Oxygen‐Barrier Coating for Long‐Term Fruit Preservation. Advanced Science, 2024, p. 2409560.
(25) Chen, C.W.; Xie, J.; Yang, F.X.; Zhang, H.L.; Xu, Z.W.; Liu, J.L.; Chen, Y.J. Development of Moisture‐Absorbing and Antioxidant Active Packaging Film Based on Poly (Vinyl Alcohol) Incorporated with Green Tea Extract and Its Effect on the Quality of Dried Eel. J. Food Process. Preserv. 2018, 42(1), e13374. doi: 10.1111/jfpp.13374.
(26) Moore, G. R. P.; Martelli, S. M.; Gandolfo, C.; do Amaral Sobral, P. J.; Laurindo, J. B. Influence of the glycerol concentration on some physical properties of feather keratin films. Food Hydrocolloids 2006, 20(7), 975-982. doi: 10.1016/j.foodhyd.2005.11.003.
(27) SHEMESH R.; KREPKER M.; GOLDMAN D.; DANIN-POLEG Y.; KASHI Y.; NITZAN N.; ... & SEGAL E. Antibacterial and antifungal LDPE films for active packaging. Polymers for Advanced Technologies ;2015 ; vol..26(1); p110-116.
(28) DE CARVALHO G.R. ; KUDAKA A.M. ; NETTO R.A. ; DELARMELINA C. ; DUARTE M.C.T. ; & LONA L.M.F. Antiviral and antibacterial activity of sodium alginate/poly (diallyldimethylammonium chloride) polyelectrolyte film for packaging applications. International journal of biological macromolecules;2023; vol..244;p125388.
(29) Li, Y., Guo, L., Yi, X., Xu, Q., Zhang, Q., Zhou, Y., ... & Chen, J. (2023). Fabrication and characterization of Plumula nelumbinis extract loaded gelatin/zein films (PNE@ GZF) to prolong strawberries shelf-life. Food Control, 154, 109989.
(30) Wu, J.; Chen, S.; Ge, S.; Miao, J.; Li, J.; Zhang, Q. Preparation, properties and antioxidant activity of an active film from silver carp (Hypophthalmichthys molitrix) skin gelatin incorporated with green tea extract. Food Hydrocolloids 2013, 32(1), 42–51. doi: 10.1016/j.foodhyd.2012.11.013.
(31) Hosseini, S.F.; Rezaei, M.; Zandi, M.; Farahmandghavi, F. Fabrication of bio-nanocomposite films based on fish gelatin reinforced with chitosan nanoparticles. Food Hydrocolloids 2015, 44, 172-182. doi: https://doi.org/10.1016/j.foodhyd.2014.09.022.
(32) JIANG H.; ZHANG W.; KHAN M.R.; AHMAD N.; RHIM J.W.; JIANG W.; & ROY S. Film Properties of Pectin Obtained from Various Fruits’(Lemon Pomelo Pitaya) Peels.. Journal of Composites Science; 2023; vol..7(9); p366.
(33) FURLAN, G. R., SILVESTRE, W. P., & BALDASSO, C. Pectin-based films with thyme essential oil: production, characterization, antimicrobial activity, and biodegradability. Polímeros, 2023, vol. 33, no. 3, e20230029.
(34) Wang, L.; Dong, Y.; Men, H.; Tong, J.; Zhou, J. Preparation and characterization of active films based on chitosan incorporated tea polyphenols. Food Hydrocolloids 2013, 32(1), 35-41. doi: 10.1016/j.foodhyd.2012.12.002.
(35) VENKATACHALAM, Karthikeyan, CHAROENPHUN, Narin, NOONIM, Paramee, et al. Influence of pomelo pericarp essential oil on the structural characteristics of gelatin-arrowroot tuber flour-based edible films. RSC advances, 2024, vol. 14, no 37, p. 27274-27287.
(36) KURTFAKI, Melike et YILDIRIM-YALCIN, Meral. Characterization of Laurus nobilis L. leaf essential oil incorporated maize starch and rice protein films. Journal of Food Measurement and Characterization, 2023, vol. 17, no 5, p. 4954-4962.
(37) Norajit, K.; Kim, K. M.; Ryu, G. H. Comparative studies on the characterization and antioxidant properties of biodegradable alginate films containing ginseng extract. Journal of Food Engineering 2010, 98(3), 377-384. doi: 10.1016/j.jfoodeng.2009.10.004.
(38) HE, Xin, LI, Min, GONG, Xuechen, et al. Biodegradable and antimicrobial CSC films containing cinnamon essential oil for preservation applications. Food Packaging and Shelf Life, 2021, vol. 29, p. 100697.
(39) KUMAR, Himanshu, AHUJA, Arihant, KADAM, Ashish A., et al. Antioxidant film based on chitosan and tulsi essential oil for food packaging. Food and Bioprocess Technology, 2023, vol. 16, no 2, p. 342-355.
(40) NISAR, M. F., ANWAR, M., & KHAN, M. I. Characteristics and antimicrobial properties of active edible films based on pectin and nanochitosan. Journal of Food Science, 2023, vol. 88, no. 2, p. 1024-1035.
(41) HAN H.S.; & SONG K.B. Antioxidant activities of mandarin (Citrus unshiu) peel pectin films containing sage (Salvia officinalis) leaf extract. International Journal of Biological Macromolecules; 2020; vol..55(5); p3173–3181.
(42) BAGHI F.; GHNIMI S.; DUMAS E.; CHIHIB N.E.; & GHARSALLAOUI A. Nanoemulsion-based multilayer films for ground beef preservation: antimicrobial activity and physicochemical properties. Molecules; 2023; vol..28(11); p4274.
(43) VAHEDIKIA N.; GARAVAND F.; TAJEDDIN B.; CACCIOTTI I.; JAFARI S.M.; OMIDI T.; & ZAHEDI Z. Biodegradable zein film composites reinforced with chitosan nanoparticles and cinnamon essential oil: Physical mechanical structural and antimicrobial attributes. Colloids and Surfaces B: Biointerfaces; 2019; vol..177; p25-32.
(44) CALO, J.R.; CRANDALL, P.G.; O’BRYAN, C.A.; RICKE, S.C. Essential oils as antimicrobials in food systems–A review. Food Control, 2015, vol. 54, p. 111–119.
(45) BHAVANIRAMYA, S.; VISHNUPRIYA, S.; AL-ABOODY M.S.; VIJAYAKUMAR R.; BASKARAN D. Role of essential oils in food safety: Antimicrobial and antioxidant applications. Grain Oil Sci. Technol., 2019 ; vol. 2 ; p. 49–55.
Round 2
Reviewer 1 Report
Comments and Suggestions for Authors
All comments have been addressed. Still measuring units have to be added to the numerical values present in the comments , even if they might be present in Table 2.
The measuring unit for DPPH antioxidant activity mentioned in the experimental section is Trolox Equivalents (TE) in millimoles, but Table 3 contains %. Perhaps Table 3 is eliminated, as both values reported are found in section 3f.
Tables 5 and 6 could be merged, without making understanding difficult.
Author Response
We have made the requested corrections

Reviewer 2 Report
Comments and Suggestions for Authors
The authors responded to my suggestions and they have addressed all the comments appropriately. In my opinion, the manuscript is now ready for acceptance.
Author Response
we have made the requested corrections
Reviewer 3 Report
Comments and Suggestions for Authors
I still do not think the manuscript can be accepted at current stage.
1 The style of references should be reversed according to the requirement of the journal.
2 The study on the effectiveness in preventing spoilage and extending shelf life in actual food products should be conducted in this paper, not in the future.
Author Response
We have made all the requested changes
